# OneActor: Consistent Subject Generation via Cluster-Conditioned Guidance

**Jiahao Wang[1,2], Caixia Yan[1,*], Haonan Lin[1], Weizhan Zhang[1,*], Mengmeng Wang[3,4],**

**Tieliang Gong[1], Guang Dai[4], Hao Sun[5]**

[1]School of Computer Science and Technology, MOEKLINNS, Xi'an Jiaotong University
[2]State Key Laboratory of Communication Content Cognition
[3]College of Computer Science and Technology, Zhejiang University of Technology
[4]SGIT AI Lab, State Grid Corporation of China
[5]China Telecom Artificial Intelligence Technology Co.Ltd

uguisu@stu.xjtu.edu.cn    {yancaixia,zhangwzh}@xjtu.edu.cn

## Abstract

Text-to-image diffusion models benefit artists with high-quality image generation. Yet their stochastic nature hinders artists from creating consistent images of the same subject. Existing methods try to tackle this challenge and generate consistent content in various ways. However, they either depend on external restricted data or require expensive tuning of the diffusion model. For this issue, we propose a novel one-shot tuning paradigm, termed OneActor. It efficiently performs consistent subject generation solely driven by prompts via a learned semantic guidance to bypass the laborious backbone tuning. We lead the way to formalize the objective of consistent subject generation from a clustering perspective, and thus design a cluster-conditioned model. To mitigate the overfitting challenge shared by one-shot tuning pipelines, we augment the tuning with auxiliary samples and devise two inference strategies: semantic interpolation and cluster guidance. These techniques are later verified to significantly improve the generation quality. Comprehensive experiments show that our method outperforms a variety of baselines with satisfactory subject consistency, superior prompt conformity as well as high image quality. Our method is capable of multi-subject generation and compatible with popular diffusion extensions. Besides, we achieve a $4\times$ faster tuning speed than tuning-based baselines and, if desired, avoid increasing the inference time. Furthermore, our method can be naturally utilized to pre-train a consistent subject generation network from scratch, which will implement this research task into more practical applications. Project page: https://johnneywang.github.io/OneActor-webpage/.

## 1 Introduction

Diffusion probabilistic models [13, 31, 34] have shown great success in image generation. As a prominent subset of them, text-to-image (T2I) diffusion models [14, 28] significantly improve artistic productivity. Designers can simply describe a subject (e.g. a character, an object, an art style) that they desire and then obtain high-quality images of the subject. However, relying on the random sampling, diffusion models fail to maintain a consistent appearance of the subject across the generated images. Taking Fig. 1(a) as an example, ordinary diffusion models denoise a random noise from the noise space to a clean latent code guided by the given prompt, where the clean code corresponds to a

---

*Corresponding authors.
This work was completed during the internship at SGIT AI Lab, State Grid Corporation of China.

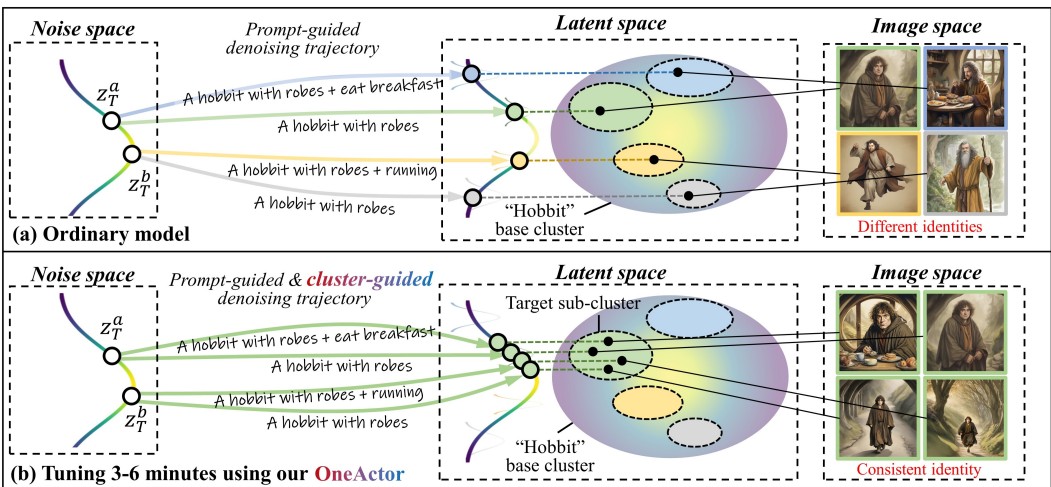

Figure 1: For every subject in the latent space, there are identity sub-clusters within the subject base cluster. (a) Given different prompts and initial noises, ordinary diffusion models generate inconsistent images from different identity sub-clusters of the "hobbit" base cluster. (b) While our OneActor, after a quick tuning, provides an extra cluster guidance and thus generates images from the same target sub-cluster that show a consistent identity. Different colors denote different identity sub-clusters.

salient image in the image space. As shown, if given 4 different prompts of the same subject "hobbit" and random noises, ordinary models will generate 4 hobbits with different identities. In other words, the generation of the ordinary models lacks *subject consistency*.

The subject consistency is a necessity in practical scenarios such as designing an animation character, advertising a product and drawing a storybook protagonist. As diffusion models prevail, many works try to harness the diffusion models to generate consistent content through the following paths. *Personalization* [8, 33, 42, 39] learns to represent a new subject from several given images and thus generates images of that. *Storybook visualization* [23, 30] manages to generate consistent characters throughout a story. However, they require external data to function, either a given image set or a specific storybook dataset. Such a requirement not only complicates practical usage, but also limits the ability to illustrating imaginary, fictional or novel subjects. More recently, *consistent subject generation* [4] is first proposed to generate consistent images of the same subject only driven by prompts. Thus, artists can easily describe the subject and start their creation, which is a more intuitive manner. Despite the pioneering work, their repetitive tuning process of the backbone model leads to expensive computation cost and possible quality degradation. Later, a tuning-free approach [37] equips the backbone model with handicraft modules to eliminate the tuning process. Yet, the extra modules double the inference time and exhibit inefficiency when generating a large number of images.

In fact, diffusion models can generate two consistent images albeit with low probability, demonstrating their inherent potential for consistent subject generation. Motivated by this, we believe all the ordinary model needs is a learned semantic guidance to tame its internal potential. To this end, we propose a novel one-shot tuning paradigm, termed as OneActor. We start from the insight that in the latent space of a pre-trained diffusion model, samples of different subjects form different clusters [4, 41], i.e. *base cluster*. In a specific base cluster, samples that share common appearance gather into the same sub-cluster, i.e. *identity sub-cluster*. In the latent space of Fig. 1, for example, a "hobbit" base cluster contains four different identity sub-clusters. Ordinary generations spread out into four different sub-clusters, causing subject inconsistency. While in our paradigm of Fig. 1(b), users first choose one satisfactory image from the generated proposals as the target. Then after a quick tuning, our OneActor quickly learns to find the denoising trajectories towards the target sub-cluster, generating images of the same subject. Throughout the process, we expect to learn in the semantic space and avoid harming the inner capacity of the latent space.

To achieve high-quality and efficient generation, we pioneer the cluster-guided formalization of the consistent subject generation task and derive a cluster-based score function. This score function underpins our novel cluster-conditioned paradigm which leverages posterior samples as cluster repre-

sentations. To systematically address overfitting, we enhance the tuning with auxiliary samples to robustly align the representation space to the semantic space. We further develop the semantic interpolation and cluster guidance scale strategies for more nuanced and controlled inference. Extensive experiments demonstrate that with superior subject consistency and prompt conformity, our method forms a new Pareto front over the baselines. Our method requires only 3-6 minutes for tuning, which is at least $4\times$ faster than tuning-based pipelines. This efficiency gain is achieved without necessarily increasing the inference time, making our method highly suitable for large-scale image generation tasks. Additionally, our method's inherent flexibility allows consistent multi-subject generation and seamless integration into various workflows like ControlNet [43].

Our main contributions are: (1) We revolutionize the consistent subject generation by pioneering the cluster-guided task formalization, which supersedes the laborious backbone tuning with a learned semantic guidance to perform efficient generation. (2) We introduce a novel cluster-conditioned generation paradigm, termed as OneActor, to pursue more generalized, nuanced and controlled consistent subject generation. It highlights an auxiliary augmentation during tuning and features the semantic interpolation and cluster guidance scale strategies. (3) Extensive experiments demonstrate our method's superior subject consistency, excellent prompt conformity and the $4\times$ faster tuning speed without inference time increase. (4) We first establish the semantic-latent guidance equivalence of T2I models, offering a promising tool for precise generation control.

## 2 Preliminaries

Before introducing our method, we first give a brief review of ordinary text-to-image diffusion models. To start with, Gaussian diffusion models [34, 13] assume a forward Markov process that gradually adds noise to normal image $\boldsymbol{x}_0$:

$$\boldsymbol{x}_t = \sqrt{\bar{\alpha}_t}\boldsymbol{x}_0 + \sqrt{1 - \bar{\alpha}_t}\epsilon, \tag{1}$$

where $t \in [0, T]$, $\epsilon \sim \mathcal{N}(\mathbf{0}, \mathbf{I})$ and $\bar{\alpha}_t$ are a set of constants. Meanwhile, a denoising network $\epsilon_{\boldsymbol{\theta}}$, usually a U-Net [32], is trained to reverse the forward process by estimating the noise given a corrupted image:

$$\mathcal{L}(\boldsymbol{\theta}) = \mathbb{E}_{t \in [1,T], \boldsymbol{x}_0, \epsilon_t} \left[ \|\epsilon_t - \epsilon_{\boldsymbol{\theta}}(\boldsymbol{x}_t, t)\|^2 \right]. \tag{2}$$

Once trained, we can sample images $\boldsymbol{x}_0$ from a Gaussian noise $\boldsymbol{x}_t$ by gradually removing the noise step by step with $\epsilon_{\boldsymbol{\theta}}$. To introduce conditional control, classifier-free guidance [14] trains $\epsilon_{\boldsymbol{\theta}}$ in both unconditional and conditional manners: $\epsilon_{\boldsymbol{\theta}}(\boldsymbol{x}_t, t, \boldsymbol{c}_\emptyset)$ and $\epsilon_{\boldsymbol{\theta}}(\boldsymbol{x}_t, t, \boldsymbol{c})$, where $\boldsymbol{c}$ is the given condition and $\boldsymbol{c}_\emptyset$ indicates no condition. Images are then sampled by the combination of two types of outputs:

$$\epsilon_{\boldsymbol{\theta}}(\boldsymbol{x}_t, t, \boldsymbol{c}_\emptyset) + s \cdot (\epsilon_{\boldsymbol{\theta}}(\boldsymbol{x}_t, t, \boldsymbol{c}) - \epsilon_{\boldsymbol{\theta}}(\boldsymbol{x}_t, t, \boldsymbol{c}_\emptyset)), \tag{3}$$

where $s$ is the guidance scale. For text control, conditions $\boldsymbol{c}$ are generated by a text encoder $E_t$, which projects text prompts $\boldsymbol{p}$ into semantic embeddings: $\boldsymbol{c} = E_t(\boldsymbol{p})$. To accelerate the pipeline, latent diffusion models [31] pre-train an autoencoder to compress images into latent codes: $\boldsymbol{z} = E_a(\boldsymbol{x})$, $\boldsymbol{x} = D_a(\boldsymbol{z})$. Thus, the whole diffusion process can be carried out in the latent space instead of the salient image space.

## 3 Method

In our task, given a user-defined description prompt $\boldsymbol{p}^{tar}$ (e.g. *a hobbit with robes*), a user-preferred image $\boldsymbol{x}^{tar}$ is generated by an ordinary diffusion model $\epsilon_{\boldsymbol{\theta}}$ and chosen as the target subject. Our goal is to equip the original $\epsilon_{\boldsymbol{\theta}}$ with a supportive network $\boldsymbol{\phi}$, formulating $\epsilon_{\boldsymbol{\theta}, \boldsymbol{\phi}}$. After a quick one-shot tuning of $\boldsymbol{\phi}$, our model can generate consistent images of the same subject with any other subject-centric prompt $\boldsymbol{p}^{sub} = \{\boldsymbol{p}^{tar}, \boldsymbol{p}\}$ (e.g. *a hobbit with robes* + *walking on the street*). To accomplish this task, We first give mathematical analysis in Sec. 3.1. Then we construct a cluster-conditioned model and tune it in a generalized manner in Sec. 3.2. During inference, we generate diverse consistent images with the semantic interpolation and cluster guidance in Sec. 3.3.

### 3.1 Derivation of Cluster-Guided Score Function

To start with, let's revisit the ordinary generation process. Given $N$ initial noises and the same subject prompt, generations of $\epsilon_{\boldsymbol{\theta}}$ fail to reach one specific sub-cluster, but spread to a base region $\mathcal{S}^{base}$ of

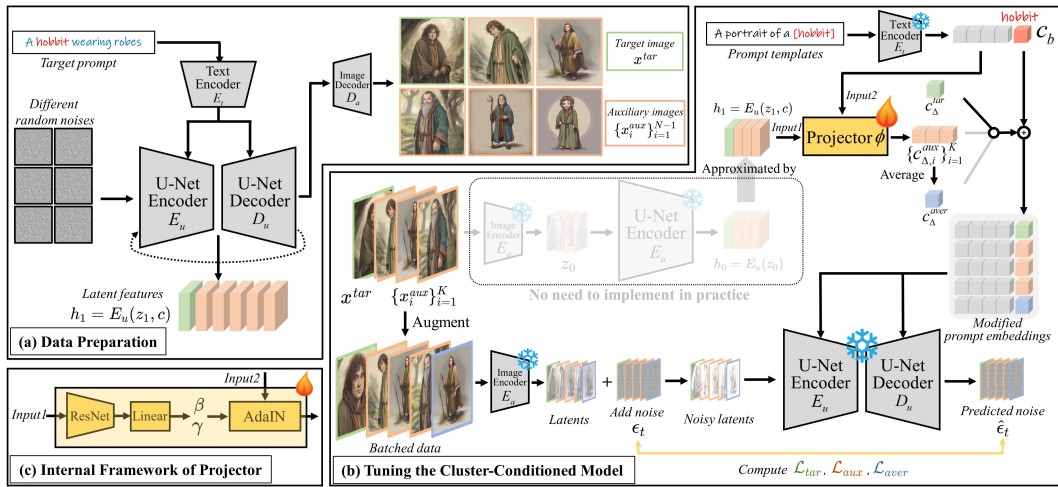

Figure 2: The overall architecture of our method. (a) We first generate base images and construct the target and auxiliary set. (b) We design a cluster-conditioned model and tune the projector with batched data. (c) The projector consists of a ResNet network, linear and AdaIN layers. Tuning and freezing weights are denoted by fire and snowflake marks. The items used to compute different objectives are outlined in different colors. The unimplemented theoretical models are semi-transparent.

different sub-clusters. If we choose one sub-cluster as the target $\mathcal{S}^{tar}$ and denote the rest as auxiliary sub-clusters $\mathcal{S}_i^{aux}$, then $\mathcal{S}^{base} = \mathcal{S}^{tar} \cup \{\mathcal{S}_i^{aux}\}_{i=1}^{N-1}$. The key to consistent subject generation is to guide the denoising trajectories of $\epsilon_{\theta}$ towards the expected target sub-cluster $\mathcal{S}^{tar}$. From a result-oriented perspective, we expect to increase the probability of generating images of the target sub-cluster $\mathcal{S}^{tar}$ and reduce that of the auxiliary sub-clusters $\mathcal{S}_i^{aux}$. Thus, if we consider the original diffusion process as a prior distribution $p(\boldsymbol{x})$, our expected distribution can be denoted as:

$$p(\boldsymbol{x}) \cdot \frac{p(\mathcal{S}^{tar} \mid \boldsymbol{x})}{\prod_{i=1}^{N-1} p(\mathcal{S}_i^{aux} \mid \boldsymbol{x})}. \tag{4}$$

We take the negative gradient of the log likelihood to derive:

$$-\nabla_{\boldsymbol{x}} \log p(\boldsymbol{x}) - \nabla_{\boldsymbol{x}} \log p(\mathcal{S}^{tar} \mid \boldsymbol{x}) + \sum_{i=1}^{N-1} \nabla_{\boldsymbol{x}} \log p(\mathcal{S}_i^{aux} \mid \boldsymbol{x}). \tag{5}$$

With the reparameterization trick of [13], we can further express the scores as the predictions of the denoising network $\boldsymbol{\theta}$ from a latent diffusion model [31] in a classifier-free manner [14]:

$$\epsilon_{\boldsymbol{\theta}}(\boldsymbol{z}_t, t) + \eta_1 \cdot \underbrace{[\epsilon_{\boldsymbol{\theta}}(\boldsymbol{z}_t, t, \mathcal{S}^{tar}) - \epsilon_{\boldsymbol{\theta}}(\boldsymbol{z}_t, t)]}_{\textit{target attraction score}} - \eta_2 \cdot \sum_{i=1}^{N-1} \underbrace{[\epsilon_{\boldsymbol{\theta}}(\boldsymbol{z}_t, t, \mathcal{S}_i^{aux}) - \epsilon_{\boldsymbol{\theta}}(\boldsymbol{z}_t, t)]}_{\textit{auxiliary exclusion score}}, \tag{6}$$

where $\eta_1, \eta_2$ are guidance control factors. If we introduce the concept of score function [35], Eq. (6) can be regarded as a combination of *target attraction score* and *auxiliary exclusion score*. This formula, termed as *cluster-guided score function*, is the core of our method. We will manage to realize it in our subsequent parts. The detailed derivation is shown in Appendix F.

## 3.2 Generalized Tuning of Cluster-Conditioned Model

In accordance with theoretical analysis of Eq. (6), we propose a novel tuning pipeline as shown in Fig. 2. In general, we incorporate the cluster representations to construct a cluster-conditioned model $\epsilon_{\boldsymbol{\theta}}(\boldsymbol{z_t}, t, \mathcal{S})$ with a supportive network $\phi$. We then prepare the data and tune the model with the help of auxiliary samples.

**Cluster-Conditioned Model.** In this model, the supportive network $\phi$ is designed to process the posterior samples into cluster representations of the semantic space. To specify, for each sample

$\boldsymbol{x}$, $\phi$ transforms its latent code $\boldsymbol{z}$ and prompt embedding $\boldsymbol{c}$ into a subject-specific vector $c_\Delta$, i.e. $c_\Delta = \phi(\boldsymbol{z}, \boldsymbol{c})$. For the framework of $\phi$, a naive way is to use a feature extractor and a space projector. Since the original U-Net encoder $E_u$ is already well-trained to extract features from the latent codes, we use it directly as the extractor. However, the U-Net extractor may cause extra computational burden. To bypass the extractor, we approximate the features of $\boldsymbol{z}_0$ with that of $\boldsymbol{z}_1$:

$$\boldsymbol{h} = E_u(\boldsymbol{z}_1, \boldsymbol{c}) \approx E_u(\boldsymbol{z}_0). \tag{7}$$

Thus, we save the intermediates $\boldsymbol{h}$ of the U-Net encoder during data generation and then directly feed them into the projector for the semantic output: $c_\Delta = \phi(\boldsymbol{h}, \boldsymbol{c}) = \phi(\boldsymbol{h}, E_t(\boldsymbol{p}))$. This approximation condenses $\phi$ to only a projector and lowers the computational cost by 30%. The output vector $c_\Delta$ represents the semantic direction of the sample's sub-cluster. We then split $\boldsymbol{c}$ into word-wise embeddings $c_i$ to locate the base word embedding $c_b$ and offset it by:

$$c_b' = c_b + c_\Delta. \tag{8}$$

The modified embedding $c_b'$ later replaces $c_b$ to form $\boldsymbol{c}'$ and guides the noise predictions. By now, all the factors (i.e. $\boldsymbol{p}, \boldsymbol{h}$) that could determine a sub-cluster are involved in the cluster-conditioned model, so we can express the cluster-related terms in Eq. (6) as:

$$\epsilon_{\boldsymbol{\theta}}(\boldsymbol{z}_t, t, \mathcal{S}) = \epsilon_{\boldsymbol{\theta},\phi}(\boldsymbol{z}_t, t, \boldsymbol{c}, c_\Delta) = \epsilon_{\boldsymbol{\theta},\phi}(\boldsymbol{z}_t, t, E_t(\boldsymbol{p}), \phi(\boldsymbol{h}, E_t(\boldsymbol{p}))). \tag{9}$$

**Generalized Tuning with Auxiliary Samples.** The major challenge of one-shot tuning is overfitting because insufficiency of data may lead to severe bias and limited diversity of generated images. To overcome this challenge, we tune the model with not only target samples, but also auxiliary samples. To elaborate, as shown in Fig. 2(a), given $\boldsymbol{p}^{tar}$ (e.g. *a **hobbit** with robes*) which contains the base word $p_i$ (e.g. **hobbit**), we first input it into the ordinary diffusion model for $N$ base images and the corresponding intermediates to form a base set: $\mathcal{X}^{base} = \{\boldsymbol{x}_i^{base}, \boldsymbol{h}_i^{base}\}_{i=1}^N$. We randomly choose one image as the target sample $\boldsymbol{x}^{tar}, \boldsymbol{h}^{tar}$ and gather the others to form an auxiliary set: $\mathcal{X}^{aux} = \{\boldsymbol{x}_i^{aux}, \boldsymbol{h}_i^{aux}\}_{i=1}^{N-1}$. We apply face crop and image flip to target image for an augmented set: $\mathcal{X}^{tar} = \{\boldsymbol{x}_i^{tar}, \boldsymbol{h}_i^{tar}\}_{i=1}^M$. Then in every tuning step, we randomly select 1 target and $K$ auxiliary samples to form a batch of data: $\mathcal{B} = \{\boldsymbol{x}^{tar}, \boldsymbol{h}^{tar}\} \cup \{\boldsymbol{x}_i^{aux}, \boldsymbol{h}_i^{aux}\}_{i=1}^K$. Thus, more data contributes to the optimization and the batch normalization can be applied to the projector, resulting in a more generalized projection. To carry out the tuning, in Fig. 2(b), we first get the latent codes by: $\boldsymbol{z} = E_a(\boldsymbol{x}), \boldsymbol{x} \in \mathcal{B}$ and add noise $\epsilon_t$ to them. Then we input the noisy latent $\boldsymbol{z}_t$, prompt $\boldsymbol{p}$ and feature $\boldsymbol{h}$ to the cluster-conditioned model with the projector $\phi$. The prompt $\boldsymbol{p}$ is a random template filled with the base word (e.g. *a portrait of a **hobbit***). We apply the standard denoising loss for both target and auxiliary samples:

$$\mathcal{L}_{tar}(\boldsymbol{\phi}) = \mathbb{E}_{t \in [1,T], \boldsymbol{z}_0^{tar}, \epsilon_t} \left[ \|\epsilon_t - \epsilon_{\boldsymbol{\theta},\phi}(\boldsymbol{z}_t^{tar}, t, E_t(\boldsymbol{p}), \phi(\boldsymbol{h}^{tar}, \boldsymbol{c}))\|^2 \right], \tag{10}$$

$$\mathcal{L}_{aux}(\boldsymbol{\phi}) = \mathbb{E}_{t \in [1,T], \boldsymbol{z}_0^{aux}, \epsilon_t} \left[ \|\epsilon_t - \epsilon_{\boldsymbol{\theta},\phi}(\boldsymbol{z}_t^{aux}, t, E_t(\boldsymbol{p}), \phi(\boldsymbol{h}^{aux}, \boldsymbol{c}))\|^2 \right]. \tag{11}$$

**Simplifying into Average Condition.** For the auxiliary exclusion score in Eq. (6), it's laborious to calculate the noise predictions of $N - 1$ auxiliary conditions in every inference step. To simplify, we substitute with an average condition $c_\Delta^{aver}$, which is derived by averaging the semantic vectors of the auxiliary instances:

$$c_\Delta^{aver} = \frac{1}{K} \sum_{i=1}^K c_{\Delta,i}^{aux} = \frac{1}{K} \sum_{i=1}^K \phi(\boldsymbol{h}_i^{aux}, \boldsymbol{c}). \tag{12}$$

Intuitively, this average condition indicates the center of all the auxiliary sub-clusters and repels the denoising trajectories away. The denoising loss is also used in this condition:

$$\mathcal{L}_{aver}(\boldsymbol{\phi}) = \mathbb{E}_{t \in [1,T], \boldsymbol{z}_0^{aver}, \epsilon_t} \left[ \|\epsilon_t - \epsilon_{\boldsymbol{\theta},\phi}(\boldsymbol{z}_t^{aver}, t, E_t(\boldsymbol{p}), c_\Delta^{aver})\|^2 \right]. \tag{13}$$

Note that the data $\boldsymbol{z}^{aver}$ is randomly chosen from target and auxiliary set. This average condition will act as the empty condition of classifier-free guidance [14] later in Sec. 3.3. During tuning, we only fire the projector from scratch and freeze all other components. As shown in Fig. 2(c), the light-weight projector comprises a ResNet [11] network, linear and AdaIN [16] layers. The complete tuning objective consists of the above 3 losses weighted by hyper-parameters $\lambda_1, \lambda_2$:

$$\mathcal{L}(\boldsymbol{\phi}) = \mathcal{L}^{tar}(\boldsymbol{\phi}) + \lambda_1 \mathcal{L}^{aux}(\boldsymbol{\phi}) + \lambda_2 \mathcal{L}^{aver}(\boldsymbol{\phi}). \tag{14}$$

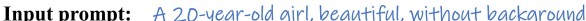

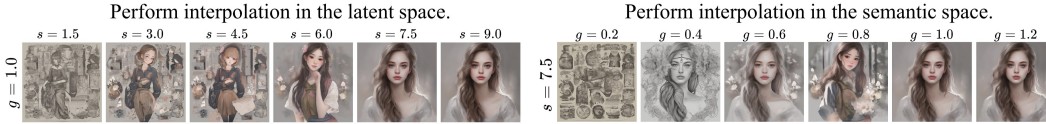

**Input prompt:** *A 20-year-old girl, beautiful, without background.*

Perform interpolation in the latent space.    Perform interpolation in the semantic space.

$s$ is the latent guidance scale, default 7.5; $g$ is the semantic interpolation scale in, default 1.0

Figure 3: The observation of semantic-latent guidance equivalence property. We vary the latent guidance scale on the left side and the semantic interpolation scale on the right, respectively. The semantic and latent manipulations show the same effect, which proves our argument.

## 3.3 Inference Strategies for Versatile Generation

In the inference stage, traditional tuning-based pipelines exhibit the imbalance between consistency and diversity and fail to extend to multiple subjects generation. To address these challenges, we propose a semantic interpolation strategy and a latent guidance scale strategy to achieve more nuanced and controlled inference. We further develop two variants to perform multiple subjects generation.

**Proof and Implementation of Semantic Interpolation.** Classifier-free guidance [14] proves that the conditional interpolation and extrapolation in the latent space, controlled by a guidance scale, reflect the same linear effect in the image space. We argue that the semantic space of the prompt embbedings also shares this property. To prove it, we carry out a toy experiment on SDXL [28]. Since the empty condition $c_\emptyset$ is also a semantic embedding obtained by encoding an empty string, we perform interpolation in the semantic space: $c' = c_\emptyset + g \cdot (c - c_\emptyset)$. Then we replace $c$ with $c'$ in the inference process to generate the images in Fig. 3. As shown, for the latent interpolation (left), when we increase the guidance scale $s$ in Eq. (3) within a proper range, the generated image will be more and more compliant with the prompt. While for the semantic interpolation (right), with $g$ increasing, we can observe the same liner reflection in the image space. This proves that the semantic space also possesses the ability of guidance interpolation. We believe that this is because the semantic and latent space are entangled by the denoising network and thus share some properties in common. With this key insight, we can precisely control the offset guidance effect of $c_\Delta$ by replacing Eq. (8) with:

$$c'_b = c_b + v \cdot c_\Delta, \tag{15}$$

where $v$ is a controllable semantic scale. We further investigate its effect in Appendix A.4. In short, a moderate $v$ leads to an optimal balance of consistency and diversity.

**Sample with Cluster-Guided Score.** Given a subject-centric prompt $p^{sub}$ in the inference stage, we first input its semantic embedding $c^{sub}$ to the tuned projector $\phi^*$ for two representations:

$$c_\Delta^{tar} = \phi^*(h^{tar}, c^{sub}), \tag{16}$$

$$c_\Delta^{aver} = \frac{1}{N-1} \sum_{i=1}^{N-1} \phi^*(h_i^{aux}, c^{sub}). \tag{17}$$

Then we approximate the N-1 auxiliary predictions in Eq. (6) with one prediction under the average condition:

$$\sum_{i=1}^{N-1} [\epsilon_{\boldsymbol{\theta},\phi^*}(\boldsymbol{z}_t, t, \boldsymbol{c}^{sub}, c_{\Delta,i}^{aux}) - \epsilon_{\boldsymbol{\theta}}(\boldsymbol{z}_t, t, \boldsymbol{c}_\emptyset)] \approx \epsilon_{\boldsymbol{\theta},\phi^*}(\boldsymbol{z}_t, t, \boldsymbol{c}^{sub}, c_\Delta^{aver}) - \epsilon_{\boldsymbol{\theta}}(\boldsymbol{z}_t, t, \boldsymbol{c}_\emptyset), \tag{18}$$

where the cluster-conditioned terms are written in the form of Eq. (9). With this approximation, Eq. (6) can be further transformed into:

$$\begin{aligned} \epsilon_{\boldsymbol{\theta}}(\boldsymbol{z}_t, t, \boldsymbol{c}_\emptyset) + \eta_1 \cdot [\epsilon_{\boldsymbol{\theta},\phi^*}(\boldsymbol{z}_t, t, \boldsymbol{c}^{sub}, c_\Delta^{tar}) - \epsilon_{\boldsymbol{\theta}}(\boldsymbol{z}_t, t, \boldsymbol{c}_\emptyset)] \\ - \eta_2 \cdot [\epsilon_{\boldsymbol{\theta},\phi^*}(\boldsymbol{z}_t, t, \boldsymbol{c}^{sub}, c_\Delta^{aver}) - \epsilon_{\boldsymbol{\theta}}(\boldsymbol{z}_t, t, \boldsymbol{c}_\emptyset)]. \end{aligned} \tag{19}$$

Now we are ready to sample consistent images with the cluster-guided score of Eq. (19). Since $\eta_1$ and $\eta_2$ play the same role as $s$ in Eq. (3), we can manipulate them for an optimal performance. For large-quantity image generation, we can set $\eta_2 = 0$ to avoid inference time increase at the expense of a slight quality decrease. Besides, since our method acts by guiding the latent code to a specific

cluster step by step, we can loosen the restrictions by applying cluster guidance to certain steps instead of all steps. The effects of these inference strategies are also discussed in Appendix A.4.

**Generating Multiple Subjects via Two Variants.** To extend the generation of single subject to multiple subjects, we develop two different variants. In the first variant, we treat the multi-subject generation as a joint distribution problem and assume that an image of two subjects belongs to the intersection sub-cluster $\mathcal{S}^{tar} = \mathcal{S}_1^{tar} \cap \mathcal{S}_2^{tar}$. Given multiple target prompts $\mathcal{P} = \{p_j^{tar}\}_{j=1}^L$ (e.g. {*a* ***hobbit*** *with robes, a white **dog**}), we combine the prompts together to one prompt $p^{tar}$ (e.g. *a **hobbit** with robes and a white **dog***) containing multiple base words. We generate multi-subject images with $p^{tar}$ and modify the projector to output multiple semantic vectors $\mathcal{C} = \{c_{\Delta,j}\}_{j=1}^L$. Thus, we can run the tuning and inference process above to generate consistent multi-subject images. However, with all subjects fixed from the beginning, this variant is unable to perform continual subject addition.

In the second variant, for each $p_j^{tar}$, we treat it as an individual and perform single-subject tuning for multiple projectors $\phi_j^*$. Then given multi-subject-centric prompt during inference, we offset the embeddings of multiple base words with the semantic outputs $\mathcal{C} = \{c_{\Delta,j}\}_{j=1}^L$ of the projectors. To avoid intersecting leakage of semantic information, we constrain the effect of every $c_{\Delta,j}$ to the subject's corresponding spatial mask in the denoising process. Since all subjects are independent in this variant, we can continually add new subjects without affecting existing ones.

## 4 Experiments

### 4.1 Baselines

To evaluate the performance of OneActor, we compare it with a wide variety of baselines: (1) tuning-based personalization pipelines, Textual Inversion (TI) [8] and DreamBooth (DB) [33] in a LoRA [15] manner; (2) encoder-based personalization pipelines, IP-Adapter (IP) [42], BLIP-Diffusion (BL) [20] and ELITE (EL) [39]; (3) consistent subject generation pipelines, TheChosenOne (TCO) [4] and ConsiStory (CS) [37]. We will denote them by the abbreviations for simplification. For personalization pipeline, we first generate one target image with the target prompt. Then we use the target image as the input to perform the personalization. Our method and the tuning-based pipeline are implemented on SDXL [28]. Note that TCO and CS have not offered official open-source codes, so we compare to the results and illustrations from their written materials for fairness. We also construct 3 ablation models: original SDXL, our model excluding $\mathcal{L}_{aver}$ and our model excluding $\mathcal{L}_{aux}, \mathcal{L}_{aver}$. More implementation details and ablation study are presented in Appendix A.3 and H.

### 4.2 Qualitative Illustration

**Single Subject Generation.** We illustrate the visual results of personalization baselines and our OneActor in Fig. 4. As shown, TI generates high-quality and diverse images but fails to maintain the consistency with the target image. With global tuning, DB shows, albeit a certain degree of consistency, limited prompt conformity. IP maintains consistency in some cases, but is unable to adhere closely to the prompt. BL suffers from inferior quality during generation. In contrast, benefiting from the intricate cluster guidance and the undamaged denoising network, our method demonstrates a balance of superior subject consistency, content diversity as well as prompt conformity. We specifically compare our OneActor with other consistent subject generation pipelines. Fig. 5 shows that all pipelines manage to generate consistent and diverse images. Yet our method preserves more consistent details (e.g. the gray shirt of the man, the green coat of the boy). Additionally, we illustrate more examples in Fig. 6, which reaffirms the excellent performance of our method.

**Multiple Subjects Generation.** We show double subjects generation comparison between baselines and the two variants of our OneActor in Fig. 7. Most pipelines, represented by DB and TCO, fail to perform multiple subjects generation. CS extends generation of single subject to multiple subjects. However, CS shows mediocre layout diversity, probably due to their unstable SDSA operations [37]. In comparison, both variants of our method excellently maintain the appearances of two subjects and display superior image diversity. In general, our method is able to generate different types of subjects including humans, animals and objects (or their combinations). Also, our method is versatile for arbitrary styles and expressions ranging from realistic to imaginary. More qualitative results including triple subjects generation and amusing applications are illustrated in Appendix B and C.

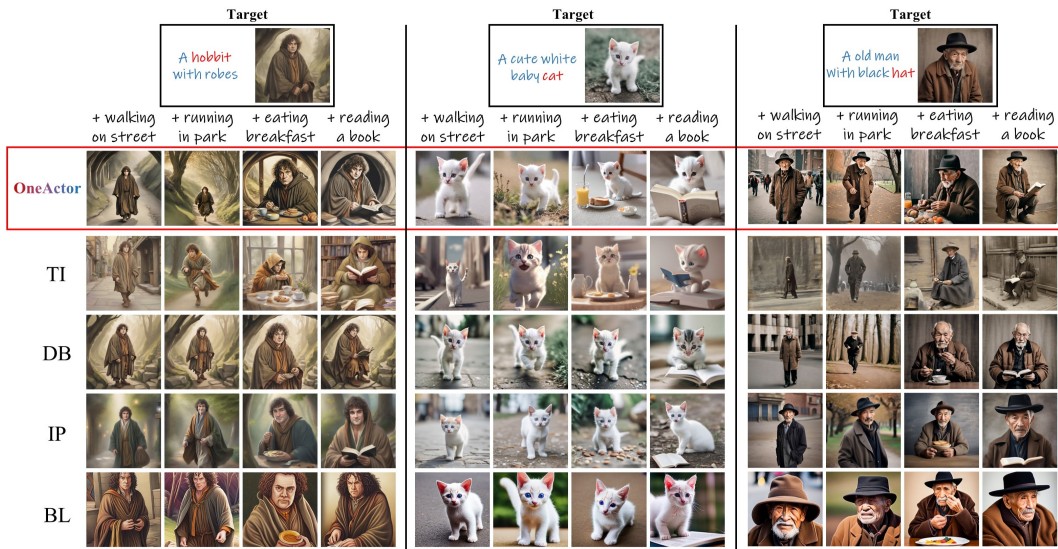

Figure 4: The qualitative comparison between personalization pipelines and our OneActor. TI lacks consistency, while DB and IP exhibit limited prompt conformity and diversity. BL suffers from poor quality in certain cases. In contrast, our method shows superior consistency, diversity as well as stability. Target prompts and base words are marked blue and red, respectively.

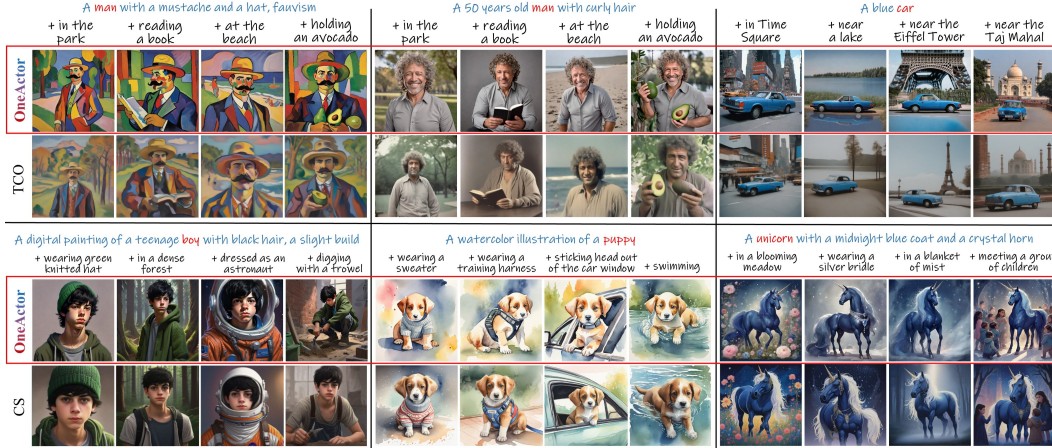

Figure 5: The qualitative comparison between consistent subject generation methods. Though all methods generate consistent images given different prompts, our OneActor refines more details such as the characters' clothes.

## 4.3 Quantitative Evaluation

**Metrics.** To provide an objective assessment, we carry out comprehensive quantitative experiments. We incorporate multi-objective optimization evaluation and encompass two objectives, identity consistency and prompt similarity, which are popular in personalization tasks [8, 33]. We instruct ChatGPT [25] to generate subject target prompts and templates. For a fair and comprehensive comparison, we adopt two metric settings from the two consistent subject generation pipelines [4, 37]. For the first setting of [4], the identity consistency is CLIP-I, the normalized cosine similarity between the CLIP [29] image embeddings of generated images and that of target image, while the prompt similarity is CLIP-T, the similarity between the CLIP text embeddings of the inference prompts and the CLIP image embeddings of the corresponding generated images. For the second setting of [37], the identity consistency is the DreamSim score [7] which focuses on the foreground subject and the prompt similarity is the CLIP-score [12].

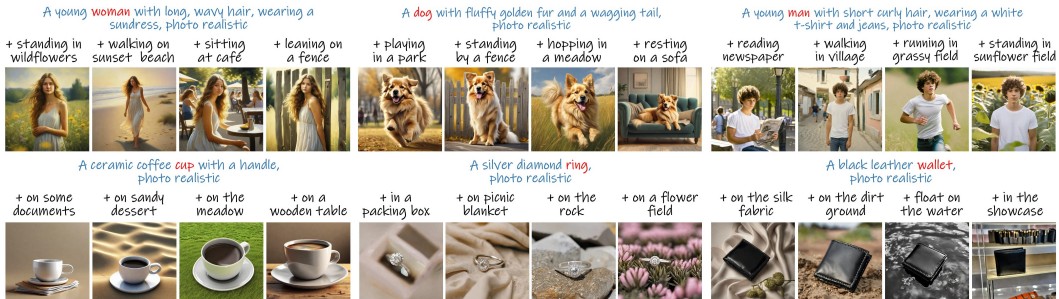

Figure 6: More qualitative results of OneActor. Our method demonstrates excellent subject consistency across characters, animals and objects.

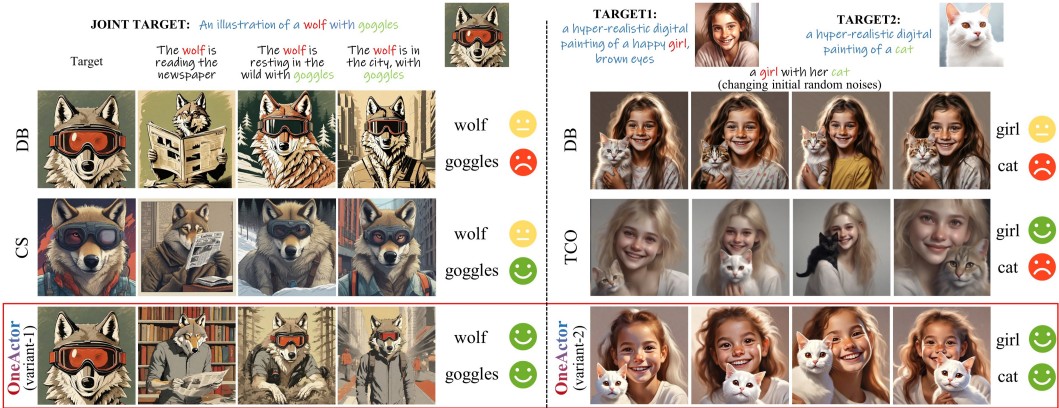

Figure 7: Double subjects generation comparison between baselines and the two variants of our OneActor. Only ConsiStory and our method are able to maintain consistency of multiple subjects.

**Results.** In Fig. 8(a), we can observe that the baselines originally form a Pareto front in the first setting. To specify, DB exhibits highest identity consistency at the cost of lowest prompt similarity and diversity. Quite the contrary, TI and BL sacrifice identity consistency for prompt similarity. TCO and IP form a moderate region with balanced identity consistency and prompt similarity. In contrast, since our method makes full use of the capacity of the original diffusion model instead of harming it, it achieves a significant boost of prompt similarity as well as satisfactory identity consistency. Note that our method Pareto-dominates IP, TI and BL. In Fig. 8(b), the original Pareto front is determined by IP and three variants of CS in the second setting. Our method pushes the front forward and shows dominance over EL, TI, DB and two variants of CS. In summary, whether in the first or second setting, our OneActor establishes a new Pareto front with a significant margin. These results correspond to the qualitative illustrations above. More experiments are shown in Appendix A.

## 5 Related Work

**Consistent Text-to-Image Generation.** A variety of works aim to overcome the randomness challenge of the diffusion models and generate consistent images of the same subject. To start with, *personalization*, given several user-provided images of one specific subject, aims to generate consistent images of that subject. Some works [8, 38] optimize a semantic token to represent the subject, while others [33, 3] tune the entire diffusion model to learn the distribution of the given subject. Later, [1, 10, 18, 36] find that tuning partial parameters of the diffusion model is effective enough. Apart from the tuning-based methods above, encoder-based methods [9, 5, 2, 42, 39] carefully encode the given subject into a representation and manage to avoid user-tuning. While *multi-concept composition* [19, 6] goes a step further to display multiple custom subjects on the same generated image via specially designed attention mechanisms. However, depending on external given images, they fail to generate imaginary or novel subjects. Recently, a new *consistent subject generation* [4] task is proposed, which aims to generate images of one subject given only its descriptive prompts.

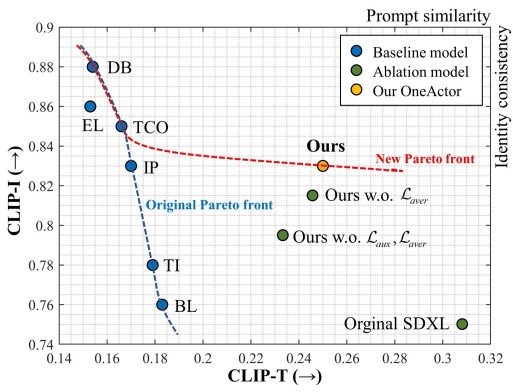

(a) Quantitative results in TheChosenOne setting.

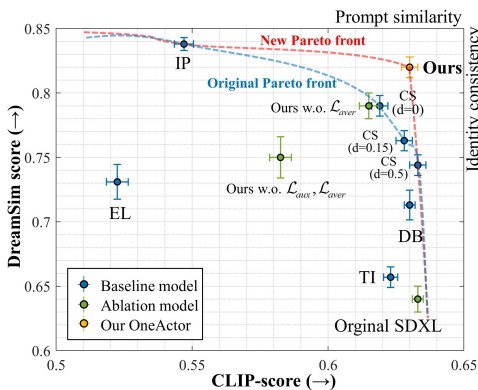

(b) Quantitative results in ConsiStory setting.

Figure 8: The quantitative comparison between baselines and our OneActor in (a) TheChosenOne setting and (b) ConsiStory setting. Either way, our method establishes a new Pareto front with superior subject consistency and prompt conformity.

Though, its tuning of the whole diffusion model is expensive and may degrade the generation quality. Later, a new approach [37] designs handicraft modules to avoid the tuning process and extends consistent generation from single subject to multiple subjects. Yet the extra modules significantly increase the inference time of every image. Besides, based on the localization of the subject, it is incapable of abstract subject generation like artistic style. For these issues, we design a new paradigm from a clustering perspective. Learning only in the semantic space, it requires shorter tuning without necessarily increasing inference time. Compared to the training-free pipelines, our pipeline can be naturally utilized to pre-train a consistent subject generation network from scratch.

**Semantic Control of Text-to-Image Generation Models.** The semantic control starts from classifier-free guidance [14], which first proposes a text-conditioned denoising network to perform text-to-image generation. It entangles the semantic and latent space so the prompt can guide the denoising trajectory towards the expected destination. Since then, many works manage to manipulate the semantic space to accomplish various tasks. For personalization, they either learn an extra semantic token to guide the latent to the subject cluster [8] or push the token embedding to its core distribution to alleviate the overfitting [45]. For image editing, they calculate a residual semantic embedding to indicate the editing direction [27, 24]. Besides, previous works [40, 21] utilize the semantic interpolation of two conditional embeddings for a mixed visual effect. These works repeatedly confirm that solely manipulating the semantic space is an effective way to harness diffusion models for various goals. Hence, in our novel cluster-guided paradigm, we transform the cluster information into a semantic representation. This representation will later guide the denoising trajectories to the corresponding cluster.

## 6 Conclusion

This paper proposes a novel one-shot tuning paradigm, OneActor, for consistent subject generation. Leveraging the derived cluster-based score function, we design a cluster-conditioned pipeline that performs a lightweight semantic search without compromising the denoising backbone. During both tuning and inference, we devise several strategies for better performance and efficiency. Extensive and comprehensive experiments demonstrate that intricate semantic guidance is sufficient to maintain superior subject consistency, as achieved by our method. In addition to excellent image quality, prompt conformity and extensibility, our method significantly improves efficiency, with an average tuning time of just 5 minutes and avoidable inference time increase. Notably, the semantic-latent guidance equivalence property proven in this paper is a potential tool for fine generation control. Furthermore, our method is also feasible to pre-train a consistent subject generation network from scratch, which implements this research task into more practical applications.

## Acknowledgements

This work was supported by the National Natural Science Foundation of China under Grant 62192781, Grant 62172326, Grant 62137002 and Grant 62302384, China Postdoctoral Science Foundation under Grant 2023M742790, Research Project Funded by the State Key Laboratory of Communication Content Cognition under Grant No. A202403, and by the Project of China Knowledge Centre for Engineering Science and Technology.

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

# Appendix

## Table of Contents

# A  Extra Experiments

## A.1  Motivation Verification

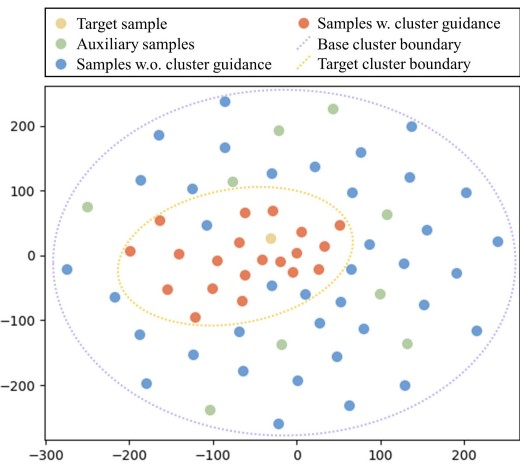

Figure 9: T-SNE illustration of the target sample, auxiliary samples, generated samples with and without our method.

Since our method is built on the clusters of the latent space, we conduct an intuitive experiment to verify our motivation. To specify, We run the whole process to generate samples, choose a target and tune the projector. Then we execute the inference twice, with and without the tuned projector. We collect the latent codes of the target sample, auxiliary samples and two rounds of generated samples. We perform t-SNE analysis to them and illustrate the results in Fig. 9. It clearly demonstrates the cluster structure in the latent space. The target (yellow) and auxiliary (green) samples together form a base cluster (the purple dotted line). Without our cluster guidance, the generated samples (blue) spread out in the base cluster region. In contrast, with our cluster guidance, the generated samples (orange) gather into a sub-cluster (the golden dotted line) centered around the target sample. Notably, some samples without cluster guidance fall into the target sub-cluster, which verifies that the original model has the potential to generate consistent images. In general, the result confirms our motivation and proves an expected function of our cluster guidance.

## A.2  Efficiency Analysis

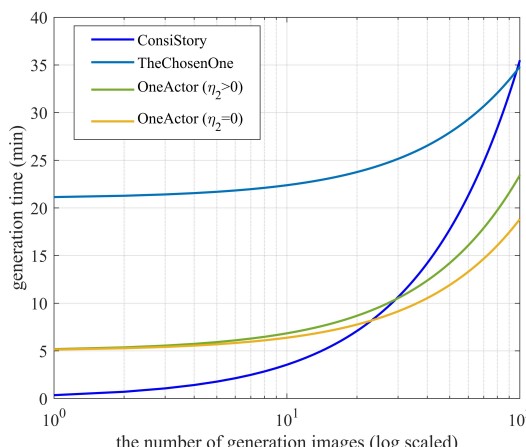

Figure 10: The latency comparison between consistent subject generation baselines and OneActor.

To evaluate the efficiency of consistent subject generation pipelines, we illustrate the generation latency of each pipeline in Fig. 10. The generation latency consists of the average tuning time and inference time. On a single NVIDIA A100, TheChosenOne [4] needs 20 minutes of tuning. After that, it takes about 8.3 seconds to generate an consistent image. ConsiStory [37] requires no tuning time, but needs about 21.3 seconds to sample one image. While our OneActor needs averagely 5 minutes of tuning. With $\eta_2 = 0$, the inference time is 8.3 seconds, which is the same as the original model. When $\eta_2 > 0$ and applying cluster guidance to steps 1-20, the inference time will rise to about 11.0 seconds. From a user's perspective, within 20 images, ConsiStory is the fastest, followed by our OneActor and then TheChosenOne. However, considering the application scenarios, the user is very likely to have a demand for large-quantity generation. In this case, our method demonstrates definite advantages. For example, when generating 100 images, ConsiStory takes about 35 minutes while our method only needs about 23 minutes. If we set $\eta_2 = 0$, with an affordable compromise of generation details, our method will reduce the total time to about 18 minutes. With the generation number increasing, our advantages will continue to grow.

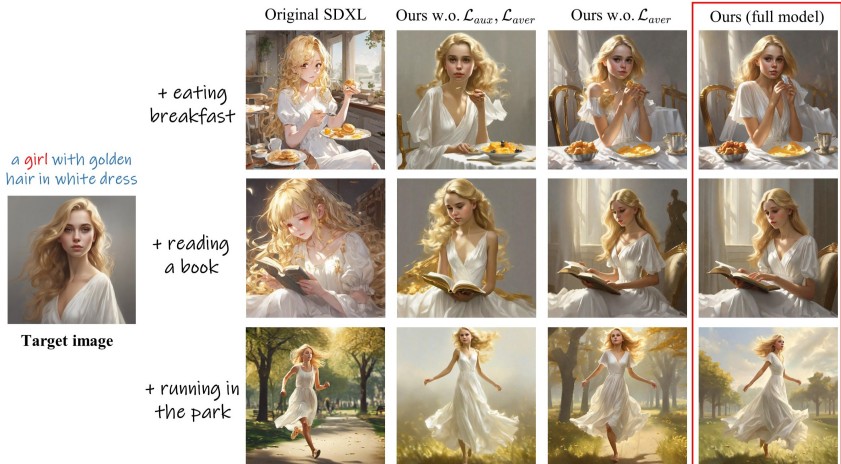

Figure 11: Illustration of the component ablation study. It shows that the proposed objective component significantly contribute to the enhanced character consistency and content diversity.

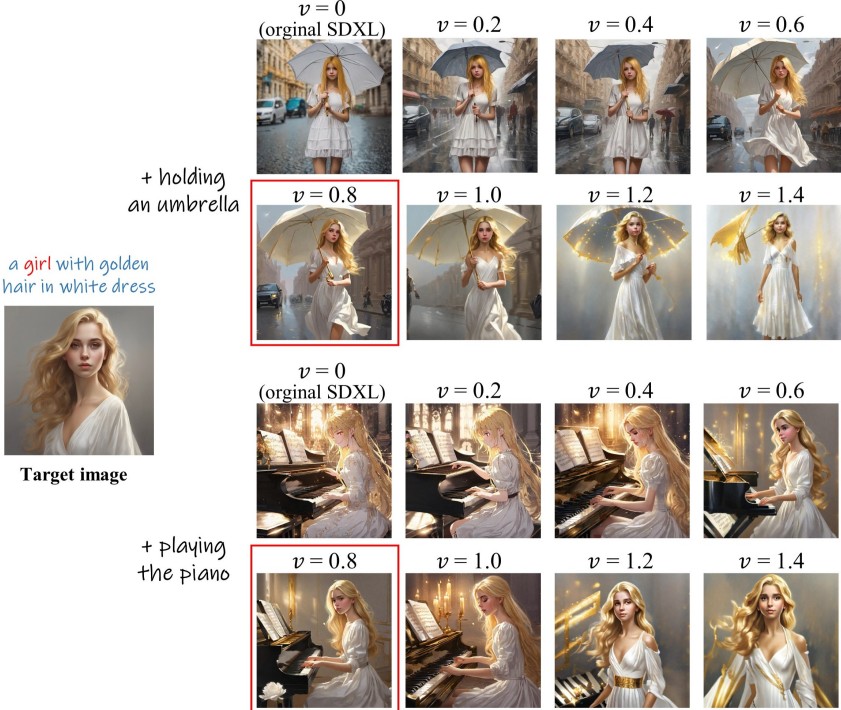

Figure 12: Illustration of the effect of semantic scale $v$. We find $v = 0.8$ is optimal because larger $v$ damages the content diversity while smaller $v$ degrades the character consistency.

## A.3  Ablation Study

The key to our tuning method lies in the three objectives in Sec. 3.2. To evaluate the effect of every functional component, we carry out a series of ablation experiments. We also construct three ablation models: original SDXL, our model excluding $\mathcal{L}_{aver}$ and our model excluding $\mathcal{L}_{aux}, \mathcal{L}_{aver}$. We run the process on these models to obtain character-centric images. For a more intuitive demonstration, we input the same initial noise and prompt to all the models and show the visual results in Fig. 11. We can observe that, first, without $\mathcal{L}_{aux}$ and $\mathcal{L}_{aver}$, the model can maintain consistency to a certain extent. However, it exhibits serious bias issue and damages the diversity of the generated images. Next, including $\mathcal{L}_{aux}$ helps stabilize the tuning and significantly improve the image quality and diversity.

Yet the identity consistency is not satisfying. Finally, in the full model, all components add up to function and generate consistent, diverse and high-quality images. The ablation experiments further validate our analysis in Sec. 3.2.

## A.4 Parameter Analysis

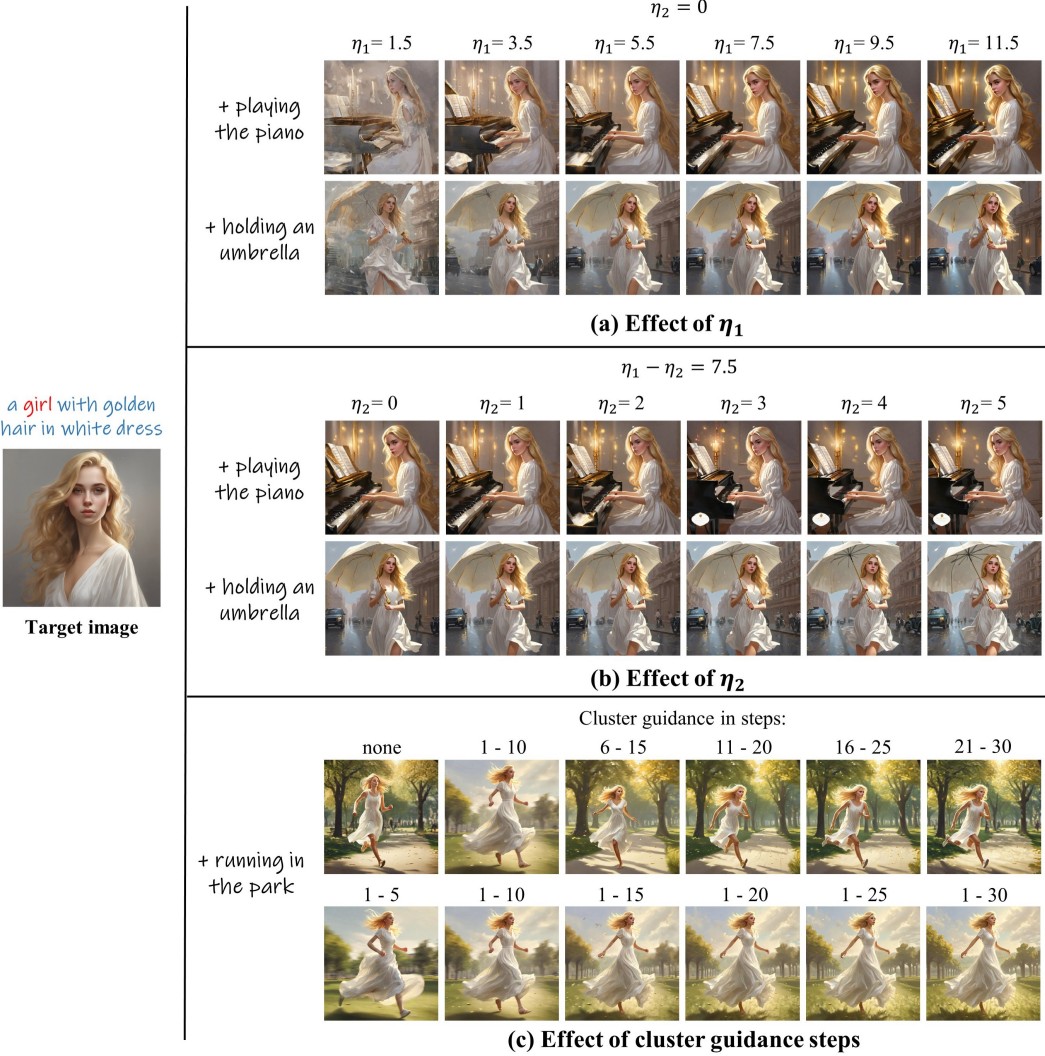

Figure 13: Illustration of parameter analysis of (a) guidance scale $\eta_1$, (b) guidance scale $\eta_2$ and (c) cluster guidance steps.

**Semantic Interpolation Scale.** We specially add semantic interpolation to enhance the flexibility and controllability during inference in Sec. 3.3. To investigate their influence on the generations, we conduct a parameter analysis of semantic scale $v$. As shown in Fig. 12, with $v$ increasing, the generated images progressively become more and more consistent with the target image. However, too large $v$ may distort the semantic embeddings and thus damage the content diversity. Hence, we set $v$ to 0.8 for a moderate performance. It's worthy noting that the results correspond with the experiments in Sec. 3.3, which proves that the interpolation is feasible to the learned $\Delta c$.

**Cluster Guidance Scale.** Since $\eta_1$ and $\eta_2$ in Eq. (19) are two crucial guidance scales for consistent subject generation, we conduct experimental analysis on them. In Fig. 13a, we set $\eta_2$ to 0 for an isolated experiment of $\eta_1$. We can observe that when $\eta_1$ increases, the image aligns more with prompt descriptions. Hence, $\eta_1$ acts like the conditional scale $s$ of the original model in Eq. (3). We use $\eta_1 = 7.5$ as the standard setting. In Fig. 13(b), we keep $\eta_1 - \eta_2 = 7.5$ and change $\eta_2$ to explore its

effect. It turns out that the performance under $\eta_2 = 0$ is already satisfying. As $\eta_2$ grows, the images exhibit more consistent details such as the facial attributes. These results align with our motivation and designs. It also proves that $\eta_1$ and $\eta_2$ can be manipulated to achieve our intended effect.

**Cluster Guidance Steps.** During inference, our cluster guidance functions like a navigator. Since the inference is step-wise, we change the inference steps to which we apply the cluster guidance, to explore the influence. Fig. 13(c) shows one interesting fact of the denoising process that, applying guidance to early steps plays a more significant role. It indicates that the early steps control the primary direction of the trajectory and determine which sub-cluster the trajectory falls in. Its influence is so deterministic that guidance in later steps can hardly change it. Hence, if not specified, we apply cluster guidance to steps 1-20.

## A.5 More Quantitative Evaluation

| Method | Subject consistency | | |
|---|---|---|---|
| | DINO-fg($\uparrow$) | CLIP-I-fg($\uparrow$) | LPIPS-fg($\uparrow$) |
| Textual Inversion | 0.452±0.130 | 0.644±0.082 | **0.322±0.061** |
| DreamBooth-LoRA | 0.735±0.078 | **0.833±0.091** | 0.296±0.064 |
| BLIP-Diffusion | 0.668±0.118 | 0.750±0.067 | 0.308±0.035 |
| IP-Adapter | 0.655±0.099 | 0.698±0.073 | 0.289±0.045 |
| OneActor(ours) | **0.756±0.073** | 0.821±0.089 | 0.299±0.057 |

Table 1: Subject consistency scores of the baselines and OneActor. The best and second-best results are marked bold and underlined, respectively.

| Method | Prompt similarity | Background diversity | | |
|---|---|---|---|---|
| | CLIP-T-score($\uparrow$) | DINO-bg($\downarrow$) | CLIP-I-bg($\downarrow$) | LPIPS-bg($\downarrow$) |
| Textual Inversion | **0.654±0.069** | **0.318±0.074** | **0.412±0.089** | 0.467±0.063 |
| DreamBooth-LoRA | 0.594±0.058 | 0.441±0.051 | 0.438±0.041 | 0.482±0.097 |
| BLIP-Diffusion | 0.547±0.074 | 0.362±0.063 | 0.446±0.064 | **0.423±0.077** |
| IP-Adapter | 0.446±0.051 | 0.388±0.054 | 0.483±0.077 | 0.494±0.101 |
| OneActor(ours) | 0.620±0.038 | 0.358±0.066 | 0.426±0.120 | 0.453±0.077 |

Table 2: Prompt similarity and background diversity scores of the baselines and OneActor.

To further evaluate the performance, we devise automatic quantitative experiments on the baselines and our OneActor. The basic settings remain the same with Sec. 4. We segment the foregrounds (fg) and backgrounds (bg) of the images by SAM [17]. For the metrics, we generally report the cosine similarity between the visual embedings of the target image and the generated images. These visual embeddings are extracted by DINO [26], CLIP [29] and LPIPS [44]. We focus on three dimensions of the metrics: (1) subject consistency: we calculate the similarity on the image foregrounds to obtain DINO-fg, CLIP-I-fg and LPIPS-fg; (2) prompt similarity: we calculate the CLIP-T-Score [12] on the whole images; (3) background diversity: we calculate the similarity on the image backgrounds to obtain DINO-bg, CLIP-I-bg and LPIPS-bg. We use the standard deviation as the error bar throughout this paper.

The results are presented in Tabs. 1 and 2. In summary, our method achieves the highest DINO-fg score and the second-highest CLIP-I-fg score, showcasing superior subject consistency. Additionally, our method ranks second across various metrics in prompt similarity and background diversity. These evaluation results highlight our method's comprehensive performance in generating consistent images.

# B Applications

Based on a mild guidance rather than a tuned backbone or attention manipulations, our OneActor can be naturally extended to various application scenarios.

## B.1 Style Transfer

For spatial location-based pipelines like ConsiStory [37], the inherent flaw is that they can't deal with abstract subjects ( e.g. art styles, brushstrokes, photo lighting). In contrast, relying on the semantic guidance, our OneActor is able to deal with any type of subject as long as it can be described. For example in Fig. 14, we perform style transfer using OneActor. To specify, we first generate an image with a target style with the prompt "*a picture in a style*" and choose **style** as the base word. Then after tuning, we generate styled images with style-centric prompts (e.g. *a fancy car in a style*).

## B.2 Storybook Generation

Since OneActor is able to perform multiple subjects generation, we can extend the ability to story visualization. As shown in Fig. 15, we first generation three target characters with three descriptive prompts. Then we perform multiple subjects generation with the second variant of our method. After tuning, the pipeline is ready to generate consistent images of one specific character or the combination of the three characters. By describing the plots with the base words, we can obtain a consistent story revolving around the three characters. Note that different from traditional story visualization pipelines, the characters in our story are arbitrary generations by diffusion models with prompts.

## B.3 Integration with Diffusion Extensions

Avoiding tuning of the denoising backbone, our method is naturally compatible with diffusion-based extensions. Here we show the integrations of StableDiffusion V1.5, StableDiffusion V2.1 and ControlNet XL in Fig. 16. The illustrations show that our method is robust across different versions of diffusion models. Besides, combined with ControlNet, our method is able to generate consistent images given different pose conditions, which highlights the compatibility of our method.

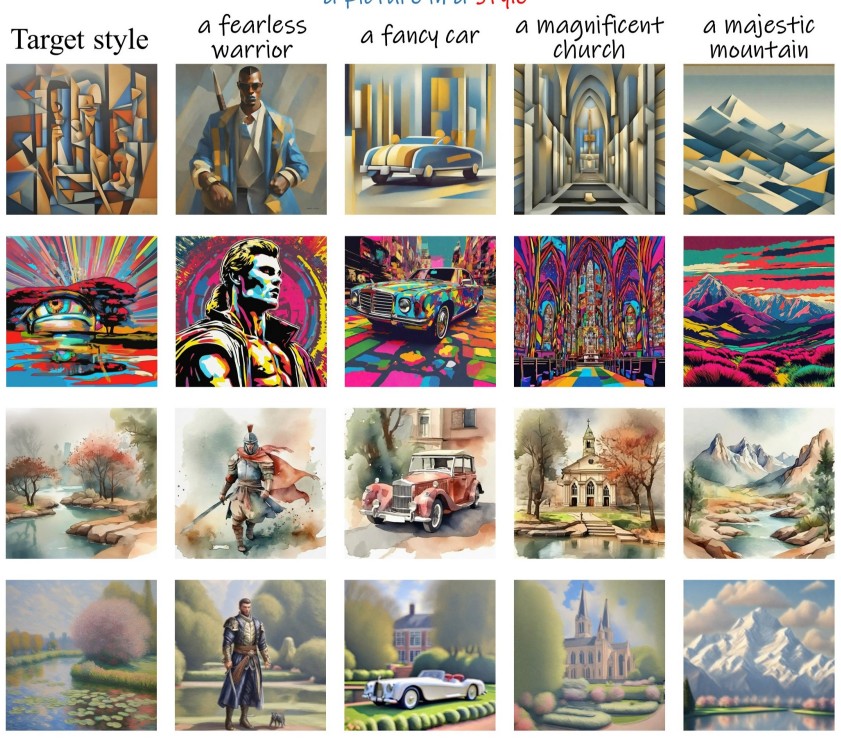

Figure 14: Style transfer using our OneActor. We generate the target image with a prompt and choose "style" as the base word. Then we perform our method to generate images in a consistent style.

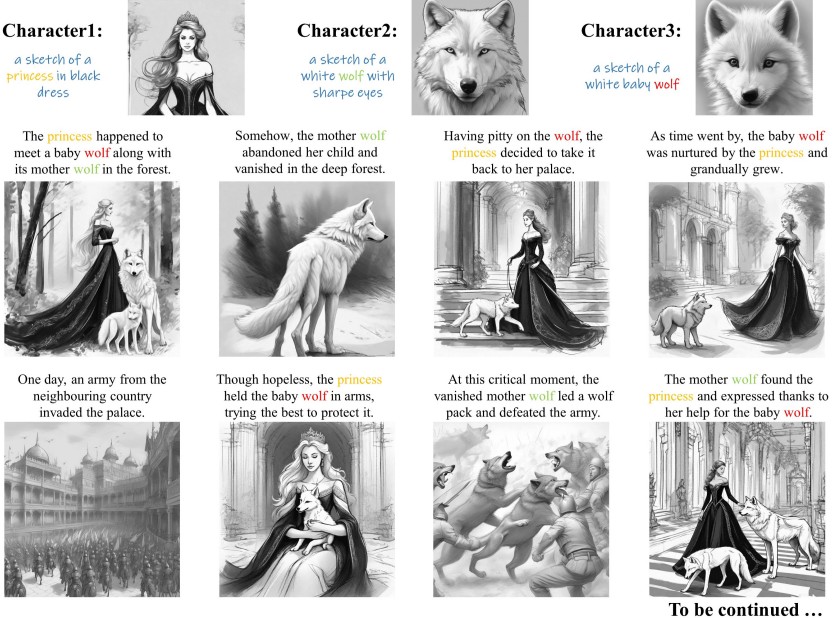

Figure 15: Story Visualization using our OneActor. We generate 3 target characters with prompts and perform the second variant of our method for multiple subjects generation. Different target characters are marked with different colors.

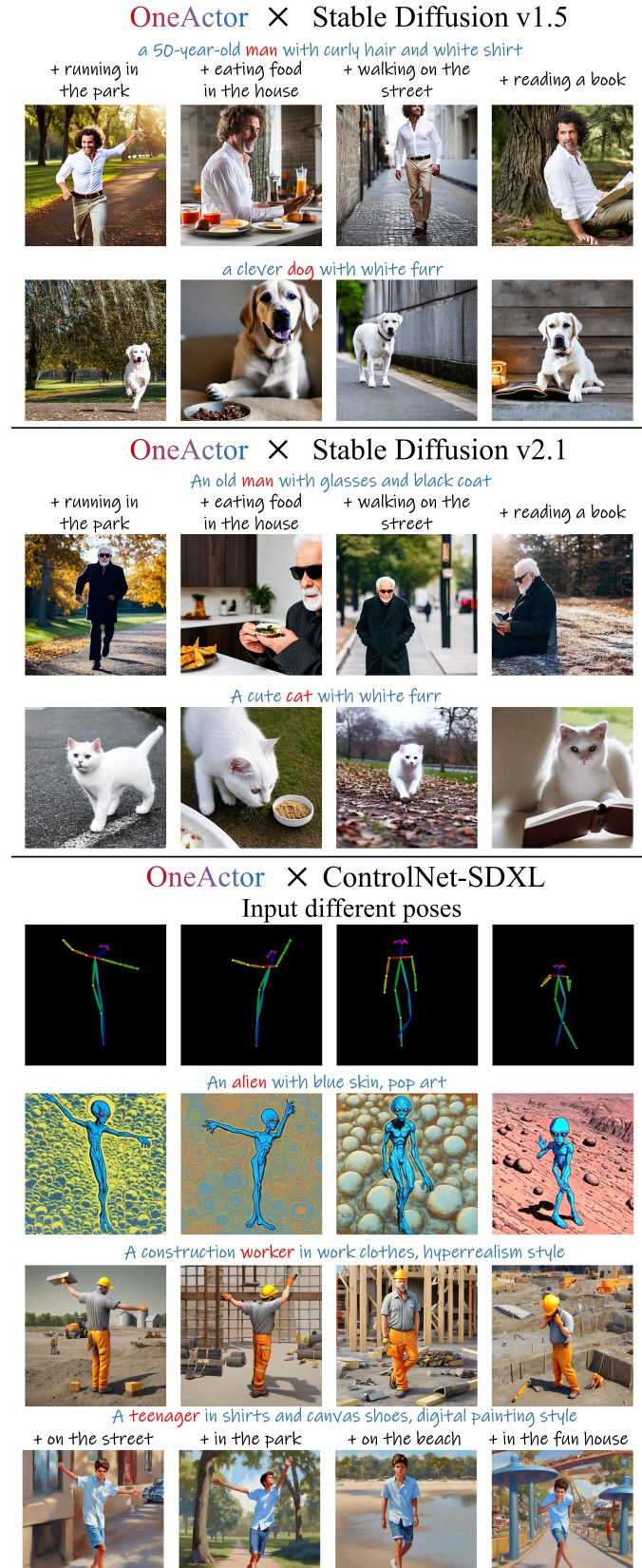

Figure 16: Integration results of our OneActor and several diffusion-based extensions. The results demonstrate the robust adaptability and compatibility of our method.

## C    More Qualitative Results

In order to comprehensively showcase the performance of our method, we provide more generation samples covering single subject, double subjects and triple subjects in Figs. 17, 18 and 19.

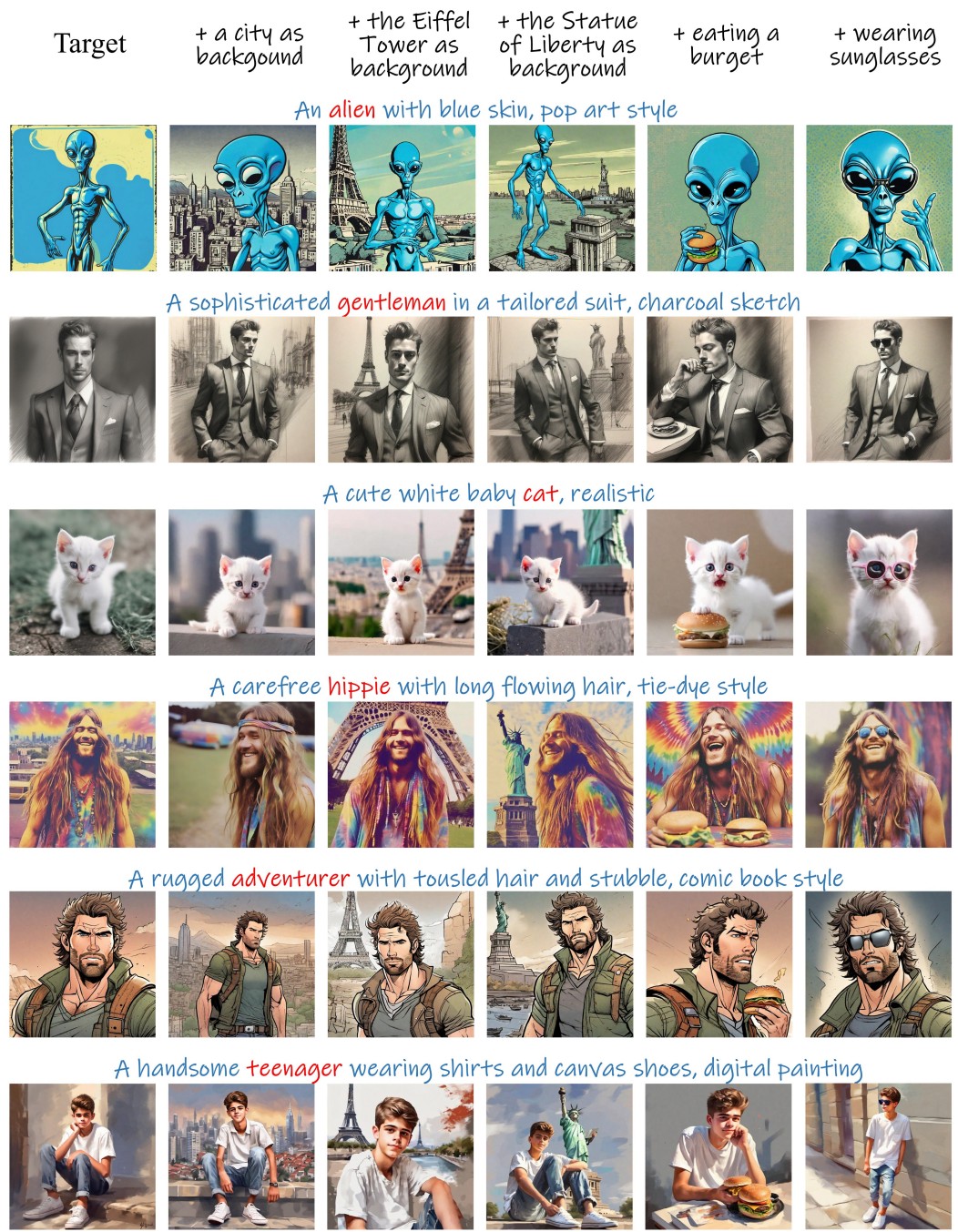

Figure 17: Illustrations of single subject generation.

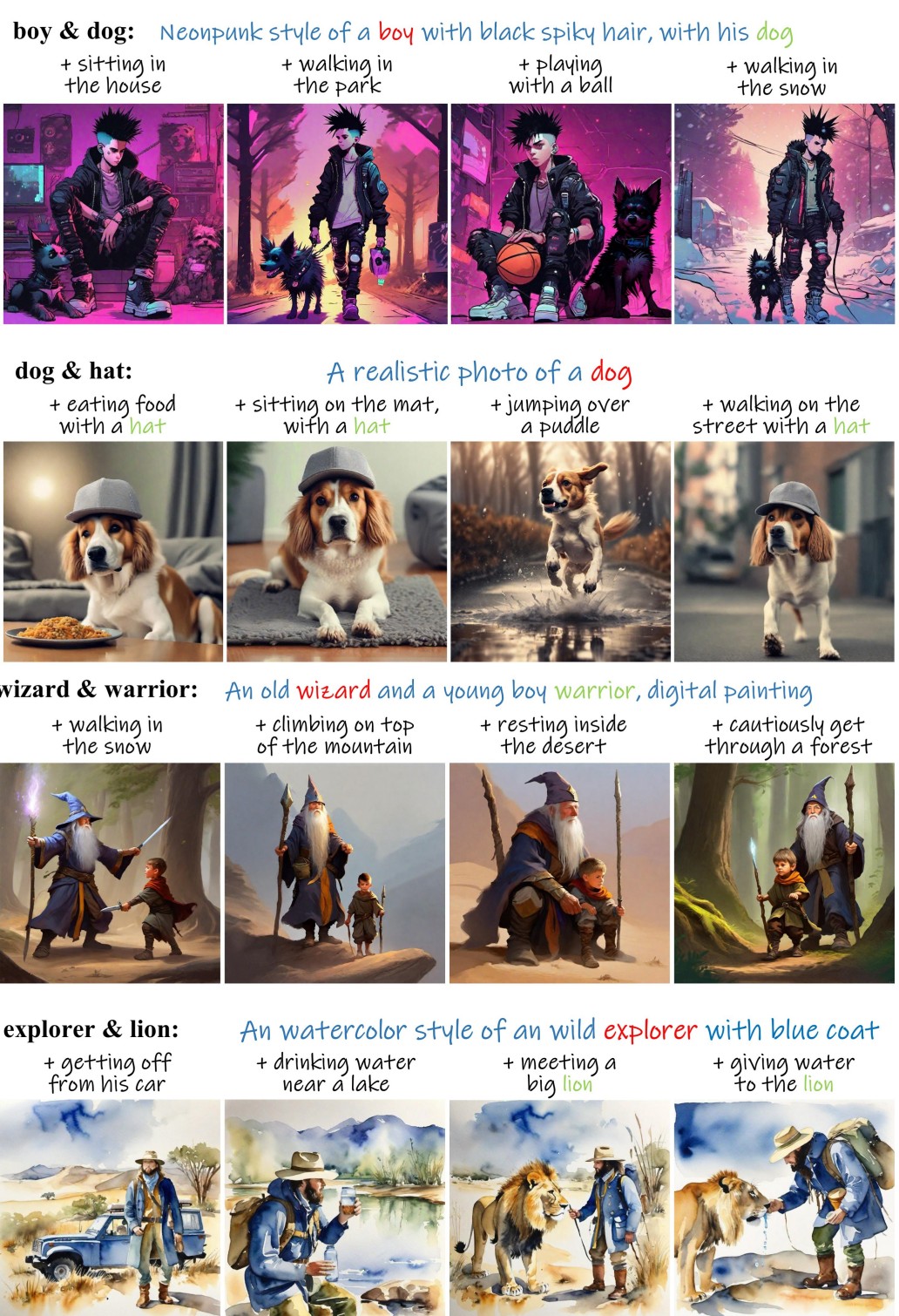

Figure 18: Illustrations of double subjects generation.

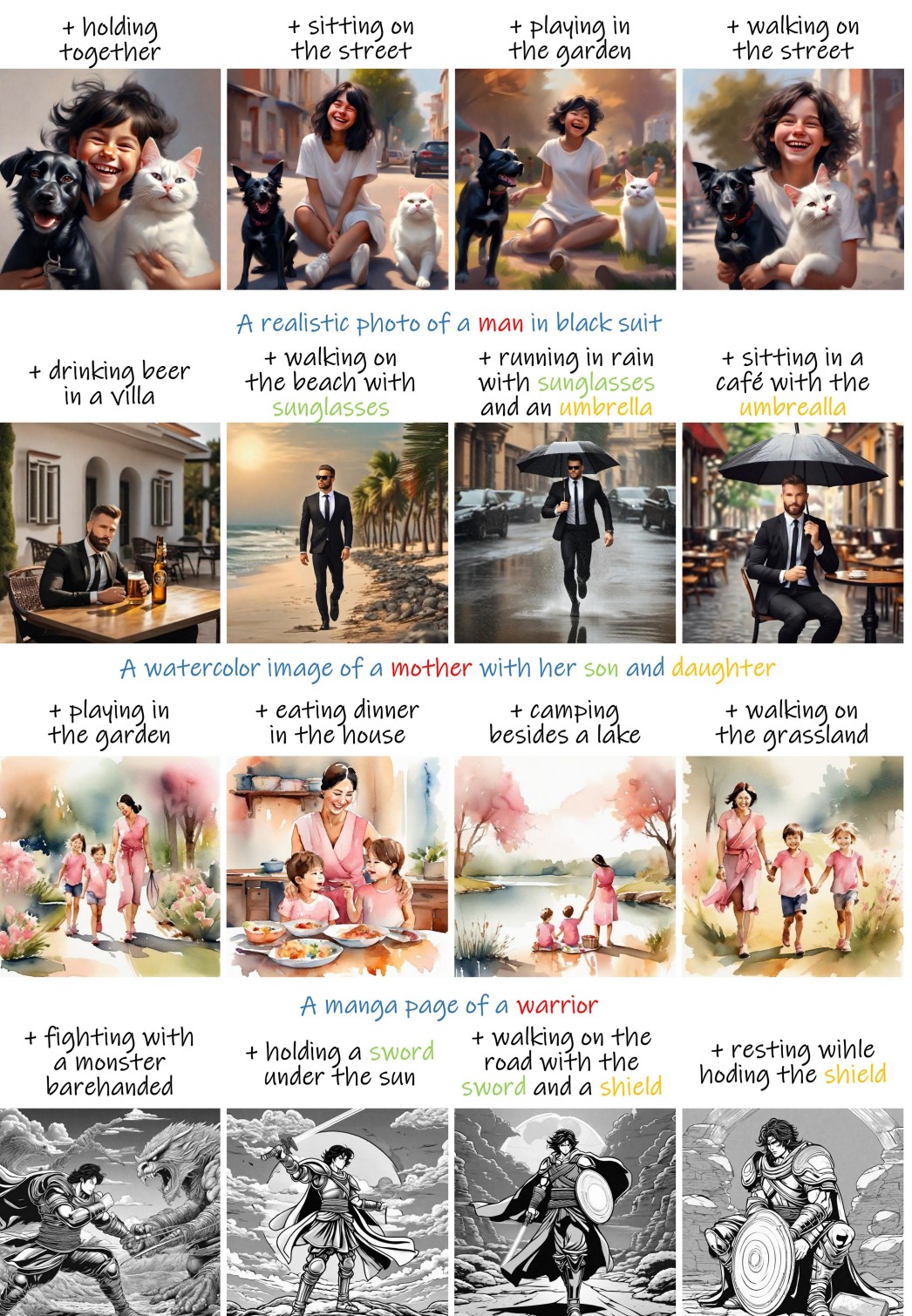

Figure 19: Illustrations of triple subjects generation.

# D    Limitations

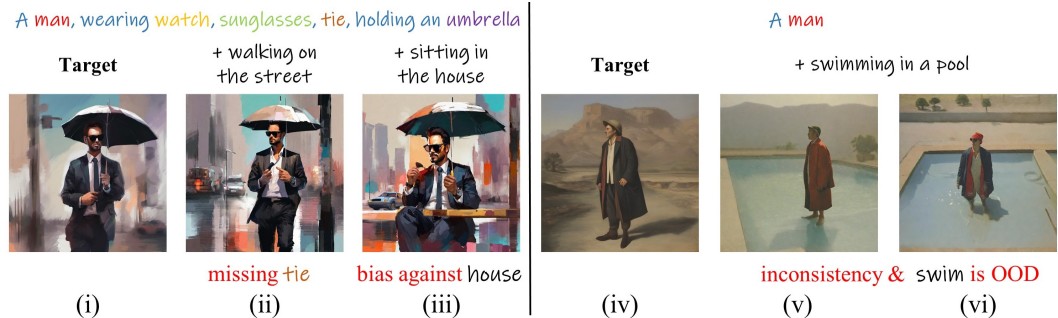

Figure 20: Limitations of OneActor. Our method struggles with generating numerous subjects on the same image and is restricted by the latent properties of diffusion models, causing bias and OOD.

Despite the above mentioned strengths, our method exhibits several limitations. One the one hand, though able to generate multiple subjects, our method struggles to simultaneously generate numerous subjects (usually $> 4$) on one image. As shown in Fig. 20(ii), in the 5 subjects generation case, the subject *tie* is neglected. On the other hand, based on a guidance manner, our method is affected by the latent properties of the original diffusion models. First, the latent clusters are entangled, which leads to inevitable bias. For the example in Fig. 20(iii), the *umbrella* causes bias against indoor environment, resulting in a mismatch with prompts. Second, as illustrated in Figs. 20(v) and (vi), if the target prompt is *a man*, its base cluster region will expand so further that it's difficult to precisely navigate to the target sub-cluster, causing inconsistency of some details. Yet, this problem can be solved by using more detailed prompts. Third, a sub-cluster distribution is rigid and incomplete, causing out of distribution (OOD) problem. In Figs. 20(vi) and (vii), *swim* is not within the scope of the target sub-cluster, which leads to mismatches with prompts. Addressing these challenging issues will be the focus of our future work.

# E    Societal Impacts

The advancements in consistent subject generation present significant societal impacts across various domains. Our work, which enhances subject consistency and efficiency in text-to-image generation, democratizes artistic creation, enabling artists and designers of all skill levels to produce consistent and high-quality visual content. It streamlines creative processes in animation, advertising and publishing, leading to substantial time and cost savings. The ability to maintain visual consistency in media and entertainment may revolutionize animation and storybook illustration.

Yet, the rapid development of creative diffusion models may lead to job displacement for artists and designers who rely on traditional methods, creating economic challenges and necessitating reskilling. Besides, ethical concerns arise around the misuse of generated content, such as the creation of deepfakes or misleading images, which can harm public trust and spread misinformation. Additionally, ensuring that generative models are trained on unbiased datasets is crucial to prevent reinforcing harmful stereotypes or biases. For these issues, since our model is based on the Stable Diffusion, we will inherit the safeguards such as the NSFW detection and the usage guideline.

# F    Detailed Derivation of Cluster-Conditioned Score Function

Given $N$ initial noises and the same subject prompt, generations of $\epsilon_{\boldsymbol{\theta}}$ fail to reach one specific sub-cluster, but spread to a base region $\mathcal{S}^{base}$ of different sub-clusters. If we choose one sub-cluster as the target $\mathcal{S}^{tar}$ and denote the rest as auxiliary sub-clusters $\mathcal{S}_i^{aux}$, then $\mathcal{S}^{base} = \mathcal{S}^{tar} \cup \{\mathcal{S}_i^{aux}\}_{i=1}^{N-1}$. The key to consistent subject generation is to guide $\epsilon_{\boldsymbol{\theta}}$ towards the expected target sub-cluster $\mathcal{S}^{tar}$. From a result-oriented perspective, we expect to increase the probability of generating images of the target sub-cluster $\mathcal{S}^{tar}$ and reduce that of the auxiliary sub-clusters $\mathcal{S}_i^{aux}$. Thus, if we consider the original

diffusion process as a prior distribution $p(\boldsymbol{x})$, our expected distribution can be denoted as:

$$p(\boldsymbol{x}) \cdot \frac{p(\mathcal{S}^{tar} \mid \boldsymbol{x})}{\prod_{i=1}^{N-1} p(\mathcal{S}_i^{aux} \mid \boldsymbol{x})}, \tag{20}$$

We take the negative gradient of the log likelihood to derive:

$$-\nabla_{\boldsymbol{x}} \log p(\boldsymbol{x}) - \nabla_{\boldsymbol{x}} \log p(\mathcal{S}^{tar} \mid \boldsymbol{x}) + \sum_{i=1}^{N-1} \nabla_{\boldsymbol{x}} \log p(\mathcal{S}_i^{aux} \mid \boldsymbol{x}). \tag{21}$$

From the score function-based perspective [35], Eq. (21) includes one unconditional score and one target conditional scores and $N-1$ auxiliary scores. With the reparameterization trick of $\boldsymbol{x}$ in terms of $\epsilon$-prediction [13]: $\boldsymbol{x_\theta}(\boldsymbol{x}_t) = (\boldsymbol{x}_t - \sigma_t \boldsymbol{\epsilon_\theta}(\boldsymbol{x}_t, t))/\alpha_t$, where $t \in [0, T]$, $\epsilon \sim \mathcal{N}(\mathbf{0}, \mathbf{I})$ and $\alpha_t, \sigma_t$ are a set of constants, the unconditional can be estimated by:

$$\boldsymbol{\epsilon_\theta}(\boldsymbol{x}_t, t) \approx -\sigma_t \nabla_{\boldsymbol{x_t}} \log p(\boldsymbol{x_t}). \tag{22}$$

We follow [14] to approximate the conditional scores in a classifier-free manner:

$$\boldsymbol{\epsilon_\theta}(\boldsymbol{x}_t, t, \mathcal{S}) - \boldsymbol{\epsilon_\theta}(\boldsymbol{x}_t, t) \approx -\eta \sigma_t \nabla_{\boldsymbol{x_t}} \log p(\mathcal{S} \mid \boldsymbol{x_t}), \tag{23}$$

where $\eta$ is a guidance control factor. Then Eq. (21) can be transformed into:

$$\boldsymbol{\epsilon_\theta}(\boldsymbol{x}_t, t) + \eta_1 \cdot [\boldsymbol{\epsilon_\theta}(\boldsymbol{x}_t, t, \mathcal{S}^{tar}) - \boldsymbol{\epsilon_\theta}(\boldsymbol{x}_t, t)] - \eta_2 \cdot \sum_{i=1}^{N-1} [\boldsymbol{\epsilon_\theta}(\boldsymbol{x}_t, t, \mathcal{S}_i^{aux}) - \boldsymbol{\epsilon_\theta}(\boldsymbol{x}_t, t)], \tag{24}$$

where $\eta_1, \eta_2$ are two scaled guidance control factors. Since we use a latent diffusion model in this paper, we transfer the above process from the image space to the latent space to derive:

$$\boldsymbol{\epsilon_\theta}(\boldsymbol{z}_t, t) + \eta_1 \cdot [\boldsymbol{\epsilon_\theta}(\boldsymbol{z}_t, t, \mathcal{S}^{tar}) - \boldsymbol{\epsilon_\theta}(\boldsymbol{z}_t, t)] - \eta_2 \cdot \sum_{i=1}^{N-1} [\boldsymbol{\epsilon_\theta}(\boldsymbol{z}_t, t, \mathcal{S}_i^{aux}) - \boldsymbol{\epsilon_\theta}(\boldsymbol{z}_t, t)]. \tag{25}$$

# G  Pseudo-Code

---

**Algorithm 1:** Tuning Process of OneActor

---

**Input** : Original diffusion model $\boldsymbol{\theta}$, projector $\phi$, target prompt $\boldsymbol{p}^{tar}$, base word $w$, prompt templates $\mathcal{P}$, max tuning step T

**Output** : Tuned projector $\phi^*$

**begin**

    Input $\boldsymbol{p}^{tar}$ into $\boldsymbol{\theta}$ for dataset $\mathcal{B} = \{\boldsymbol{z}^{tar}, \boldsymbol{h}^{tar}\} \cup \{\boldsymbol{z}_i^{aux}, \boldsymbol{h}_i^{aux}\}_{i=1}^K$

    **for** $s = 1$ *to* $T$ **do**

        Randomly choose $\boldsymbol{p} \in \mathcal{P}$ and fill it with $w$

        Obtain the embedding $\boldsymbol{c}$ of $\boldsymbol{p}$ using the text encoder of $\boldsymbol{\theta}$

        **for** $\boldsymbol{z}_0$ *in* $\{\boldsymbol{z}^{tar}, \boldsymbol{z}_i^{aux}\}$ **do**

            $\boldsymbol{z}_t \leftarrow \sqrt{\bar{\alpha}_t} \boldsymbol{z}_0 + \sqrt{1 - \bar{\alpha}_t} \epsilon$, where $t \in [0, T]$, $\epsilon \sim \mathcal{N}(\mathbf{0}, \mathbf{I})$ and $\bar{\alpha}_t$ are constants

        **end**

        $c_\Delta^{tar} \leftarrow \phi(\boldsymbol{h}^{tar}, \boldsymbol{c})$

        $\{c_{\Delta,i}^{aux}\}_{i=1}^K \leftarrow \{\phi(\boldsymbol{h}^{tar}, \boldsymbol{c})\}_{i=1}^K$

        $c_\Delta^{aver} \leftarrow \frac{1}{K} \sum_{i=1}^K c_{\Delta,i}^{aux}$

        Calculate loss $\mathcal{L}(\phi)$ using Eqs. (10)-(14)

        $\mathcal{L}(\phi).\texttt{backward}()$

    **end**

**end**

---

# H Experiment Details

## H.1 Implement Details

**General Settings.** We implement our method based on SDXL, which is a common choice by most of the related works. All experiments are finished on a single NVIDIA A100 GPU. All images are generated in 30 steps. During tuning, we generate $N = 11$ base images. We use $K = 3$ auxiliary images each batch. The projector consists of a 5-layer ResNet, 1 linear layer and 1 AdaIN layer. The weight hyper-parameters $\lambda_1$ and $\lambda_2$ are set to 0.5 and 0.2. We use the default AdamW optimizer with a learning rate of 0.0001, weight decay of 0.01. We tune with a convergence criterion, which takes 3-6 minutes in most cases. During inference, we set the semantic interpolation scale $v$ to 0.8 if not specified. We set the cluster guidance scale $\eta_1$ and $\eta_2$ to 8.5 and 1.0. We apply cluster guidance to the first 20 inference steps and normal guidance to the last 10 steps.

**Multi-Word Subject.** Since we apply semantic offset to the base word of the target prompt, a multi-word subject (e.g. *pencil box*) may cause problem. For this case, our method automatically adapts the projector to output multiple offsets for the words, while other parts remain the same.

**Subject Mask Extraction.** When generating multiple subjects, our method perform mask extraction to the involved subjects. To specify, in each generation step, we collect all the cross-attention maps that correspond to the token of each subject. We apply Otsu's method to the average of the maps for a binary mask for each subject.

## H.2 Packages License

- SDXL [28] implementation at
  https://huggingface.co/stabilityai/stable-diffusion-xl-base-1.0.
- DreamBooth-LoRA [33] and Textual Inversion [8] implementations on SDXL at
  https://huggingface.co/diffusers.
- ELITE [39] implementation at
  https://github.com/csyxwei/ELITE.
- BLIP-diffusion [20] implementation at
  https://huggingface.co/salesforce/blipdiffusion/tree/main.
- IP-Adapater [42] implementation at
  https://github.com/tencent-ailab/IP-Adapter.
- CLIP [29], DINO [26] and SAM [17] implementation at
  https://github.com/huggingface/transformers.
- LPIPS [44] implementation at
  https://github.com/richzhang/PerceptualSimilarity.

## H.3 Prompts for Evaluation

Target prompts, where the base words are denoted bold:

- A sophisticated **gentleman** in a tailored suit, charcoal sketch.
- A graceful **ballerina** in a flowing tutu, watercolor.
- A carefree **hippie** with long flowing hair, tie-dye style.
- A wise **elder** with a wrinkled face and kind eyes, pastel portrait.
- A rugged **adventurer** with tousled hair and stubble, comic book style.
- A trendy **hipster** with thick-rimmed glasses and a beanie, street art.
- A spirited **athlete** in mid-stride, captured in a dynamic sculpture.
- A curious **child** with wide eyes and messy hair, captured in a whimsical caricature.
- A playful **panda** with black patches and fluffy ears, cartoon animation.
- A majestic **tiger** with orange stripes and piercing gaze, photorealistic illustration.

- A graceful **deer** with antlers and doe eyes, minimalist vector art.
- A curious **raccoon** with bandit mask and bushy tail, pastel-colored sticker.
- A wise **owl** with feathers and intense stare, ink brushstroke drawing.
- A sleek **dolphin** with gray skin and playful smile, digital pixel art.
- A mischievous **monkey** with long tail and banana, clay sculpture.
- A colorful **parrot** with vibrant feathers and beak, paper cut-out animation.
- A cuddly **koala** with fuzzy ears and button nose, felt fabric plushie.
- A majestic **eagle** with sharp talons and soaring wings, metallic sculpture.
- A noble **griffin** in a tapestry, its presence commanding attention with intricate detail and grandeur.
- A mischievous **imp** captured in charcoal, its sly grin dancing off the page.
- A radiant **unicorn** in stained glass, casting prisms of light with its ethereal beauty.
- An ominous **kraken** in sculpture, its tentacles frozen in time, threatening the bravest souls.
- A resplendent **phoenix** in metalwork, its fiery wings ablaze with eternal flames.
- An enigmatic **sphinx** in relief, guarding ancient secrets with stoic grace.
- A whimsical **faun** in woodcarving, its playful spirit captured in intricate detail.
- A formidable **werewolf** in digital art, its primal fury palpable yet contained.
- A mystical **mermaid** in underwater mural, her siren song echoing through ocean depths.
- A legendary **dragon** in castle tapestry, its scales shimmering with timeless majesty and power.
- A majestic **lion** in bronze sculpture, its mane flowing with regal power.
- A stealthy **ninja** in ink drawing, moving silently through the shadows.
- A serene **monk** in oil painting, meditating amidst tranquil surroundings.
- A futuristic **robot** in 3D rendering, with sleek lines and glowing eyes.

Subject-centric prompts, where $[b]$ indicates the base word:

- $[b]$, at the beach.
- $[b]$, in the jungle.
- $[b]$, in the snow.
- $[b]$, in the street.
- $[b]$, with a city in the background.
- $[b]$, with a mountain in the background.
- $[b]$, with the Eiffel Tower in the background.
- $[b]$, near the Statue of Liberty.
- $[b]$, near the Sydney Opera House.
- $[b]$, floating on top of water.
- $[b]$, eating a burger.
- $[b]$, drinking a beer.
- $[b]$, wearing a blue hat.
- $[b]$, wearing sunglasses.
- $[b]$, playing with a ball.
- $[b]$, as a police officer.
- $[b]$, on a rooftop overlooking a city skyline.
- $[b]$, in a library surrounded by books.
- $[b]$, on a farm with animals in the background.

- [b], in a futuristic cityscape.
- [b], on a skateboard performing tricks.
- [b], in a traditional Japanese garden.
- [b], in a medieval castle courtyard.
- [b], in a science laboratory conducting experiments.
- [b], in a yoga studio practicing poses.
- [b], at a carnival with colorful rides.
- [b], in a cozy coffee shop sipping a latte.
- [b], in a sports stadium cheering for a team.
- [b], in a space station observing Earth.
- [b], in a submarine exploring the depths of the ocean.
- [b], on a film set directing a movie scene.
- [b], in an art gallery admiring paintings.
- [b], in a theater rehearsing for a play.
- [b], at a protest holding a sign.
- [b], in a classroom teaching students.
- [b], in a space observatory studying the stars.
- [b], walking on the street.
- [b], eating breakfast in the house.
- [b], running in the park.
- [b], reading a book in the library.

