# OpenReview forum: "OneActor: Consistent Subject Generation via Cluster-Conditioned Guidance"
_NeurIPS.cc/2024/Conference — NeurIPS 2024 poster_

### Official Review · Reviewer_BPQh · 2024-06-24

**Soundness:** 3
**Presentation:** 3
**Contribution:** 3
**Rating:** 6
**Confidence:** 5

**Summary:**

The authors present a generation paradigm called “OneActor” to generate consistent subject in text-to-image generating tasks. The core of this algorithm is called as “cluster-guided score function”, which is based on the concept of score function and created to maintain the consistency of generated images. Additionally, the superiorities of this method for consistency performance and faster tuning are shown quantitatively and qualitatively through various experiments.

**Strengths:**

1.	The authors creatively present an insight that samples of different subjects form different clusters, and analyze it in detail, which is the inspiration of their method.

2.	The derivation of formulas is in complete detail without errors and codes are committed.

3.	The experiments are sufficient and solidly conducted, including both qualitative and quantitative comparisons. The ablation studies are sensible and include user studies as well.

**Weaknesses:**

The overall quality of the paper is quite good, but some problems still exist:

1. The core of this method seems to split the scores of CFG into a target and an auxiliary part for customized generation and maintain consistent subject, which is also derived in detail in Appendix F. The novelty is straightforward, and it would be better to compare the results with the research of multi-concept text-to-image tasks, for example:

Kumari, Nupur, et al. "Multi-concept customization of text-to-image diffusion." Proceedings of the IEEE/CVF Conference on Computer Vision and Pattern Recognition. 2023.

2. There are some typos. For instance, as shown in line 119, the conditional and unconditional scores have the same notation ϵ_θ (x_t,t,c_∅ ).

3. Some experimental results are not good enough. For example, the beard of an old man shown in Figure.4 is not consistent. Additionally, a hobbit generated by DB look better than the “OneActor”.

**Questions:**

1. The authors present a framework to maintain consistency property. How to connect the insights shown in this paper with score-sde? In other words, can we use score-matching to understand clusters?

2. As shown in Eq.(5), since it is an expectation why is the expression not as follows?
p(x)∙p(S^tar |x)/(∏_(i=1)^(N-1) p(S_i^aux |x)+p(S^tar |x) )

3. Since the average condition indicates the center of all the auxiliary sub-clusters, is there a strategy to make sure a conception like “radius” for the clusters?

4. What are the advantages over methods based on the attention layer of neural networks?

**Limitations:**

1.	As shown in weakness, the method cannot capture all the details of a given target image.

2.	The subject-centric shown in this paper can be described by a word so that it would be meaningful to further explore more complex and diverse cases.

---

> ### Author Rebuttal · Authors · 2024-08-05
>
> We greatly appreciate your insightful review of our work. For the weakness and questions, we will response to each of them individually.
> ***
> ## Weakness 1: Comparison with multi-concept customization pipeline
> We provide a qualitative comparison with FreeCustom[1], the current state-of-the-art multi-concept customization method, in the Figure 22 of the global rebuttal pdf. Note that FreeCustom is officially implemented on StableDiffusion V1.5 and thus exhibits averagely lower image quality. In terms of subject consistency and prompt conformity, which are backbone-irrelevant, our method still outperforms FreeCustom. This reaffirms the effectiveness of the proposed cluster guidance method. Due to the time limit, we are not able to perform more comprehensive experiments of multi-concept customization methods. We will add these methods to the Related Work section and add more experiments in the new version of our paper.
>
> [1] FreeCustom: Tuning-Free Customized Image Generation for Multi-Concept Composition, CVPR 2024
> ## Weakness 2: Typo
> We will carefully review our paper sentence-by-sentence and try to correct every type error. Thanks for your meticulous review!
> ## Weakness 3: Imperfect consistency
> For the "hobbit" case comparison between DB and our method, solely from the perspective of subject consistency, DB indeed beats our method. Yet we believe our method achieves a more balanced performance of subject consistency, background and layout diversity. We believe this balanced performance is more meaningful in the consistent subject generation task. Due to the non-ideal distribution of the clusters, there will be some inconsistent minor details such as the beard or clothing decorations. Addressing this challenge will be the focus of our future work.
> ## Question 1: Connection with score-SDE
> In our opinion, the basic logic of our method is the same as that of the score-SDE. The methodology of score-SDE [2] is to transform the log-likelihood into the predicted score of the denoising network in a score-matching manner. We are inspired and employ the transformation in the derivation from Equation (5) to Equation (6) in our submission. The training of $\epsilon_{\boldsymbol \theta,\boldsymbol \phi^*}(\boldsymbol z_t,t,\boldsymbol c^{sub},c_\Delta^{tar})$ and $\epsilon_{\boldsymbol \theta,\boldsymbol \phi^*}(\boldsymbol z_t,t,\boldsymbol c^{sub},c_\Delta^{aver})$ are essentially two score-matching processes of the target sub-cluster score $\nabla_{\boldsymbol{x}}\log p(\mathcal{S}^{tar}\mid\boldsymbol{x})$ and the auxiliary sub-cluster score $\sum_{i=1}^{N-1}\nabla_{\boldsymbol{x}}\log p(\mathcal{S}_i^{aux}\mid\boldsymbol{x})$.
>
> [2] Score-based generative modeling through stochastic differential equations
> ## Question 2: Formula expression
> There are many different ways to express the expectation of the task including $p(\boldsymbol{x})\cdot\frac{p(\mathcal{S}^{tar}\mid\boldsymbol{x})}{\prod_{i=1}^{N-1}p(\mathcal{S}_i^{aux}\mid\boldsymbol{x})+p(\mathcal{S}^{tar}\mid\boldsymbol{x})}$. We eventually adopted the simplified Equation (5) for the following reasons:
> - Retaining only the multiplication and division terms simplifies the subsequent formula transformation.
> - $p(\boldsymbol{x})\cdot\frac{p(\mathcal{S}^{tar}\mid\boldsymbol{x})}{\prod_{i=1}^{N-1}p(\mathcal{S}_i^{aux}\mid\boldsymbol{x})+p(\mathcal{S}^{tar}\mid\boldsymbol{x})}$ can be transformed into
>
>    $p(\boldsymbol x)\cdot\frac{1}{\prod_{i=1}^{N-1}p(\mathcal{S}_i^{aux}\mid\boldsymbol x)/p(\mathcal{S}^{tar}\mid\boldsymbol x)+1}$.
>
>    Thus, the core expectation is also to increase $\frac{p(\mathcal{S}^{tar}\mid\boldsymbol x)}{\prod_{i=1}^{N-1}p(\mathcal{S}_i^{aux}\mid\boldsymbol x)}$ as the Equation (5) does.
> - During implementation, the target and auxiliary samples all contribute to the average condition, which completes the center of the whole cluster. So the simplification is more like a theoretical trick.
> ## Question 3: Radius concept
> We believe there exists the concept of *radius* in the latent space, yet it’s hard to precisely determine the *radius*. Because if we consider the $(\boldsymbol z_t,t)$ as a whole, the latent space is complicated and might not be a Euclidean space. The *radius* might appear to be a geometric manifold that extends continuously over $t$. It will be a interesting yet challenging future topic to develop a strategy to determine the *radius*.
> ## Question 4: Advantages over attention manipulation methods
> Compared with the training-free pipelines based on attention manipulation, our tuning-based pipeline has the following advantages:
> - Our method can be naturally utilized to pre-train a consistent subject generation network from scratch, which implements this research task into more practical applications.
> - Based on the stable tuning process of an auxiliary network rather than the U-Net backbone, our method demonstrates more robust performance across different settings. By contrast, the attention manipulation to the U-Net backbone utilized by training-free baselines exhibits occasional degradations of the image quality (see the first and fourth images of the ‘*man*’ case generated by StoryDiffusion in the Figure 21 of the global rebuttal pdf; also see the first image of the ‘*girl-cat*’ case generated by FreeCustom in the Figure 22).
> - Though our method takes averagely 5 minutes to tune, it doesn’t increase the inference time as training-free baselines do. Thus, as the generation number increases, our method outperforms the training-free methods in terms of total generation time (see the Table 3 in the global rebuttal pdf).
> ***
> We believe the rebuttal stage is highly meaningful, and we will add the rebuttal discussion to the new version of our paper as much as possible. If there is any omission in our response or if you have any additional question, please feel free to let us know. Thanks again for your great efforts!

---

> > ### Comment · Reviewer_BPQh · 2024-08-11
> > **comment**
> >
> > I think the responses of the authors have addressed my concerns, and the submitted experimental results reflect the effectiveness of the method. I will not reduce my score.

---

> > > ### Author Response · Authors · 2024-08-11
> > >
> > > Dear Reviewer BPQh,
> > >
> > > We greatly appreciate your satisfaction with our responses and will make revisions to our work based on your comments in order to further improve the quality of our work.
> > >
> > > Thanks again for your valuable suggestions and comments. We enjoy communicating with you and appreciate your efforts!

---

### Official Review · Reviewer_7cPE · 2024-07-10

**Soundness:** 2
**Presentation:** 3
**Contribution:** 2
**Rating:** 5
**Confidence:** 4

**Summary:**

This Study proposes a one-shot tuning paradigm for efficient and consistent subject generation. They introduce two inference strategies to mitigate overfitting, improving generation quality significantly. The semantic space of diffusion mode trained under the proposed method has the same interpolation property as the latent space, offering enhanced control over generation. Comprehensive quantitative and qualitative experiments prove that the method is effective.

**Strengths:**

1. The motivation and the methods proposed in the entire article are justifiable. The illustrations are incredibly exquisite, and the presented results effectively prove the efficacy of their method
2. For practitioners in the field of AIGC, I indeed believe it holds practical value.

**Weaknesses:**

Some of the related techniques aren't as novel as the author argued. Such as “We are the first to prove that the semantic space of the diffusion model has the same interpolation property as the latent space does.”. As far as I know, it is quite common to do identity mixing[1,2,3,4] in the semantic space in human photo customization. [3,4] also use  two inference strategies including classifier-free guidance and semantic interpolation, which is the same as the method in the paper

1. FaceStudio: Put Your Face Everywhere in Seconds
2. PhotoMaker: Customizing Realistic Human Photos via Stacked ID Embedding
3. InstantID: Zero-shot Identity-Preserving Generation in Seconds
4. FlashFace: Human Image Personalization with High-fidelity Identity Preservation

**Questions:**

1. The cases provided in the paper are relatively simple. Can the algorithm handle more complex multi-entity scenes, such as those involving more than one person or cat?
2. How does this compare to a seemingly simpler approach [1]?

[1] StoryDiffusion: Consistent Self-Attention for Long-Range Image and Video Generation

---

> ### Author Rebuttal · Authors · 2024-08-05
>
> We sincerely thank you for your valuable reviews. For the weakness and questions, we will response to each of them individually.
> ***
> ## Weakness: Novelty of semantic interpolation
> We believe our discovery is fundamentally different from previous works. Ordinary diffusion process can be denoted as $p(\boldsymbol{z}\mid \boldsymbol{c},s)$, where $\boldsymbol{c}$ is the semantic condition and $s$ is the weight of classifier-free guidance score. Previous works [1,2,3,4] prove that the semantic interpolation $\boldsymbol c'=\boldsymbol c_1+\alpha\cdot(\boldsymbol c_2-\boldsymbol c_1)$ between two conditional embeddings $\boldsymbol c_1$ and $\boldsymbol c_2$ shows a mixed visual effect and a gradual visual change, which can be written as: $p(\boldsymbol z\mid\boldsymbol c’,s)\approx p(\boldsymbol z\mid\boldsymbol c_1,s)\cdot p(\boldsymbol z\mid\boldsymbol c_2,s)$. This discovery is utilized to mix different input conditions for a fancy hybrid output image. By contrast, our contribution lies in proving that the semantic interpolation $\boldsymbol c'=\boldsymbol c_\emptyset+\alpha\cdot(\boldsymbol c_1-\boldsymbol c_\emptyset)$ between the unconditional embedding $\boldsymbol c_\emptyset$ and conditional embedding $\boldsymbol c_1$ has the same effect with the guidance interpolation $s'=0+\beta\cdot(s_1-0)$ between no guidance $s=0$ and standard guidance scale $s=s_1$. The above conclusion can be written as: $p(\boldsymbol z\mid \boldsymbol c',s_1)\approx p(\boldsymbol z\mid \boldsymbol c_1,s')$. It indicates that semantic manipulation has the same effect with the sampling guidance strength, which we believe is essentially different from the conclusion of previous works. Yet we apologize for the misinformation caused by our inappropriate wording and we will polish our phrasing to be more precise and objective. We will also add the above-mentioned works to the Related Work section in the new version of our paper. Thanks so much for your nuanced review!
>
> [1] FaceStudio: Put Your Face Everywhere in Seconds
>
> [2] PhotoMaker: Customizing Realistic Human Photos via Stacked ID Embedding
>
> [3] InstantID: Zero-shot Identity-Preserving Generation in Seconds
>
> [4] FlashFace: Human Image Personalization with High-fidelity Identity Preservation
> ## Question 1: Multi-entity scenes
> We provided illustrations of double subjects generation and triple subjects generation in Figures 17 and 18 in the Appendix C of the submission. The results show that our method is able to deal with the complex 2-entity and 3-entity scenarios. The storybook generation in the Appendix B.2 also proves that our method is capable of complex sequential generation. Yet when dealing with more numerous subjects (usually >4), as discussed in the Appendix D, some subjects might be neglected. We also observe this defect in the original SDXL model and addressing this foundational limitation will be the focus of our future work.
> ## Question 2: Comparison with StoryDiffusion
> We carry out qualitative comparison with the contemporaneous work StoryDiffusion [5] in Figures 20 and 21 in the global rebuttal pdf. Generally, we think our method performs better in terms of subject consistency and background diversity. In a broader context, compared with the training-free pipelines like StoryDiffusion, our tuning-based pipeline has the following advantages:
> - Our method can be naturally utilized to pre-train a consistent subject generation network from scratch, which implements this research task into more practical applications.
> - Based on the stable tuning process of an auxiliary network rather than the U-Net backbone, our method demonstrates more robust performance across different settings. By contrast, the attention manipulation to the U-Net backbone utilized by training-free baselines exhibits occasional degradations of the image quality (see the first and fourth images of the ‘*man*’ case generated by StoryDiffusion in the Figure 21 of the global rebuttal pdf).
> - Though our method takes averagely 5 minutes to tune, it doesn’t increase the inference time as training-free baselines do. Thus, as the generation number increases, our method outperforms the training-free methods in terms of total generation time (see the Table 3 in the global rebuttal pdf).
>
> [5] StoryDiffusion: Consistent Self-Attention for Long-Range Image and Video Generation
> ***
> We believe the rebuttal stage is highly meaningful, and we will add the rebuttal discussion to the new version of our paper as much as possible. If there is any omission in our response or if you have any additional question, please feel free to let us know. Thanks again for your great efforts!

---

> ### Author Response · Authors · 2024-08-12
>
> Dear Reviewer 7cPE,
>
> We sincerely thank you for your valuable reviews. To each of the question, we have provided detailed respones in the rebuttal stage, which we hope could address your concerns. If you have any further questions or there is any omission in our responses, please feel free to let us know. And we would be extremely grateful if you could consider increasing the rating after reviewing our responses.
>
> Thanks again for your great efforts!

---

### Official Review · Reviewer_SGyo · 2024-07-11

**Soundness:** 3
**Presentation:** 3
**Contribution:** 3
**Rating:** 5
**Confidence:** 3

**Summary:**

This paper proposes OneActor, a one-shot tuning paradigm for consistent subject generation in text-to-image diffusion models, driven solely by prompts and utilizing learned semantic guidance to bypass extensive backbone tuning.

A cluster-conditioned model is introduced to formalize consistent subject generation from a clustering perspective. To mitigate overfitting, the tuning process is augmented with auxiliary samples and employs two inference strategies: semantic interpolation and cluster guidance, significantly enhancing generation quality.

Comprehensive experiments demonstrate that this method outperforms various baselines, providing superior subject consistency, prompt conformity, and high image quality. OneActor supports multi-subject generation, is compatible with popular diffusion extensions, offers faster tuning speeds, and can avoid increased inference time. Additionally, this paper proves that the semantic space of the diffusion model has the same interpolation properties as the latent space, presenting a promising tool for fine generation control.

**Strengths:**

- The problem of changing the text prompt while preserving the ID is important.
- Learning semantic guidance to harness its internal potential is insightful.
- The auxiliary augmentation in one-shot tuning is novel in text-to-image applications.
- The code is provided.
- Quantitative results are clearly presented in Figure 7.
- According to Figure 6, this paper also considers multiple subjects.

**Weaknesses:**

- All demos are in cartoon or artistic styles; no photorealistic results are shown. The objects are limited, raising questions about the fairness of the quantitative results.
- The CLIP score may be insufficient to measure ID similarity.
- Figure 2 is somewhat cluttered. Please focus on highlighting the most important technical contribution.
- No qualitative ablation study is provided.
- Efficiency is a concern. Few quantitative results on efficiency are provided. A processing time of 3-6 minutes may be too long for a text-to-image pipeline.

**Questions:**

Please refer to the weakness.

**Limitations:**

Well discussed.

---

> ### Author Rebuttal · Authors · 2024-08-04
>
> We sincerely thank you for your valuable reviews. For the weakness and questions, we will response to each of them individually.
> ***
> ## Weakness 1: Limited styles and objects in the main part of the paper
> The 9-page main part of the submission is limited in space so we placed the most crucial comparisons with the baselines in it and placed more visual results in the Appendix of the submission. Because the baselines (TheChosenOne and ConsiStory) have not yet open-sourced their codes, we can only compare to the results of their written materials, which accidentally limits the variety of the style and central subject in the main part. We showed a little more results of photorealistic styles and objects in the Appendix. B and C. Additionally, we provide more visual results concentrating on objects and photorealistic style in Figure 21 of the global rebuttal pdf. The results demonstrate that our method performs well in this setting.
> ## Weakness 2: CLIP score is not insufficient
> CLIP-I-score is the metric that the baselines utilize so we followed and reported it in the main part of our submission. Yet we completely agree that CLIP-I score is insufficient to measure the subject ID. Thus in addition, we provided a more comprehensive quantitative evaluation in the Appendix A.5, where we reported DINO score and LPIPS score as well. Beside these common metrics, we also notice that recently some human-aligned benchmarks are proposed to specially evaluate the image consistency, such as DreamBench++[1]. Due to the time limit, we are not able to perform comprehensive experiments on such benchmarks in the rebuttal stage, but we will try to add such experiments in the new version of our paper.
>
> [1] DreamBench++: A Human-Aligned Benchmark for Personalized Image Generation, arXiv2406.16855
> ## Weakness 3: Cluttered figure
> We will revise the Figure 2 to highlight the projector and the cluster guidance methodology. Thanks for your valuable advice!
> ## Weakness 4: Qualitative ablation study
> Since there is no space in the main text, we provided the illustration of component ablation study in the Appendix A.3 and parameter analysis in the Appendix A.4. Please refer to it for further review.
> ## Weakness 5: Quantitative efficiency results
> We provided an efficiency analysis in the Appendix A.2 and add an efficiency statistics in Table 3 of the rebuttal pdf. Though our method takes averagely 5 minutes to tune, it doesn’t increase the inference time as training-free baselines do. From the user’s perspective, when generating <20 images for one subject from scratch, ConsiStory is the fastest, followed by our method and then TheChosenOne. However, considering the application scenarios, the user is very likely to have a demand for large-quantity generation (>50 images). In this case, our method demonstrates definite advantages. For example, when generating 100 images, ConsiStory takes about 35 minutes while our method only needs about 23 minutes in total. If we set $\eta_2=0$, with an affordable compromise of generation details, our method will reduce the total time to about 18 minutes. With the generation number increasing, our advantages will continue to grow.
> ***
> We believe the rebuttal stage is highly meaningful, and we will add the rebuttal discussion to the new version of our paper as much as possible. If there is any omission in our response or if you have any additional question, please feel free to let us know. Thanks again for your great efforts!

---

> ### Author Response · Authors · 2024-08-12
>
> Dear Reviewer SGyo,
>
> We sincerely thank you for your valuable reviews. To each of the question, we have provided detailed respones in the rebuttal stage, which we hope could address your concerns. If you have any further questions or there is any omission in our responses, please feel free to let us know. And we would be extremely grateful if you could consider increasing the rating after reviewing our responses.
>
> Thanks again for your great efforts!

---

> > ### Comment · Reviewer_SGyo · 2024-08-13
> >
> > I continue to observe a data bias in this method, as most examples are in cartoon or artistic styles. The new examples provided by the author have intensified my concerns. Although the newly added examples feature real objects, they still exhibit a strong artistic influence (coffee, dog). This application is important, especially in real-world scenarios, but I believe the current presentation of the paper remains somewhat skewed. The author's explanation of limited space doesn't convince me. If a method is truly general, it should ideally demonstrate its effectiveness across a wider range of domains. I will keep my score unchanged temporarily, but I think we need to discuss the practical significance of this paper.

---

> > > ### Author Response · Authors · 2024-08-13
> > >
> > > Thank you so much for your further review and we would like to offer some clarifications regarding your concern.
> > >
> > > Our method is a **one-shot tuning** pipeline based on the pre-trained text-to-image generation model StableDiffusionXL (SDXL). The whole procedure involves:
> > > 1) generating an **initial target image using SDXL**;
> > > 2) performing our one-shot tuning pipeline to **produce consistent images according to the target image**.
> > >
> > > In each set of examples shown in our submission and rebuttal, the first image is the target generated by SDXL. For instance, in the 'dog' example in the rebuttal pdf (Fig. 21 middle top), the image of 'playing in a park' is generated by the base model SDXL. Our method then performs one-shot tuning based on this image to produce the subsequent images of 'standing by a fence', 'hopping in a meadow', and 'resting on a sofa'.
> > >
> > > **What our method achieves is to maintain excellent consistency with the target image. The artistic influence you observed in images like the 'dog' and 'coffee cup' is due to the training bias of the SDXL model, rather than a limitation of our method.**  During our experiments, we also observed that SDXL often produces animation-like images even when we use prompts like 'photorealistic', 'a photo of', and 'photography'.
> > >
> > > In fact, when SDXL does produce truly photorealistic images like the ***'ring in a packing box', 'wallet on the silk fabric',*** and ***'man reading newspaper'*** in the rebuttal pdf (Fig. 21), **our method effectively maintains this photorealistic style**. According to the rebuttal rules, we are not able to provide additional illustrations, but we do believe the current qualitative examples sufficiently demonstrate both the theoretical contributions and practical values of our method.
> > >
> > > We sincerely hope this response helps address your concerns. Thank you again for your thoughtful feedback and your efforts!

---

### Official Review · Reviewer_iFKj · 2024-07-11

**Soundness:** 4
**Presentation:** 4
**Contribution:** 4
**Rating:** 7
**Confidence:** 4

**Summary:**

This paper proposes to formalize the consistent content generation problem from a clustering perspective. By designing a one-shot tuning paradigm with a cluster-conditioned model, the proposed pipeline OneActor can achieve faster tuning while maintaining superior subject consistency. Extensive experiments show that the semantic interpoloation and cluster guidance can contribute to the high quality of consistent subject generation.

**Strengths:**

1. The formalization of consistent subject generation from a clustering perspective is novel and elegant.
2. This one-shot tuning paradigm with cluster guidance is creative and could pave a new path of fine control with diffusion models.
3. Experiments are extensive and solid, showing the superior performance compared to baseline methods.
4. This paper is well organized and of high quality.

**Weaknesses:**

There is a typo in line 119 for the unconditional manner being repeated twice.

**Questions:**

How do you design the projector network? What's the motivation behind the current design?

**Limitations:**

The authors have adequately addressed the limitation and social impact in Appendix D and E, respectively.

---

> ### Author Rebuttal · Authors · 2024-08-04
>
> We greatly appreciate your commendation of our work. It is without doubt a tremendous encouragement for us. For the weakness and question, we will response to each of them individually.
> ***
> ## Weakness: Typo
> We will carefully review our paper sentence-by-sentence and try to correct every type error. Thanks for your meticulous review!
> ## Question: The design of the projector network
> This is a very meaningful question which we've discussed a lot ourselves. In our pipeline, the role of the projector network is to transform a token embedding to another adjusted token embedding with respect to the visual feature. Its essence is a transformation inside one modality (semantic) conditioned on another modality (visual). It's similar to the style transfer task, which transfers one image style to another style with respect to the image elements. Inspired by this, we utilize a projector based on AdaIN, which is a popular transformation layer in the style transfer task. Then the ResNet and linear layer are naturally chosen to extract the visual feature and output the $\beta$ and $\gamma$ of the AdaIN layer. According to the experiments, our design functions well in the one-shot tuning setting. Furthermore, our method is potential to be utilized to pre-train a consistent subject generation model from scratch. In thus setting, we do believe there might be a more perfect network design for the projector.
> ***
> We believe the rebuttal stage is highly meaningful, and we will add the rebuttal discussion to the new version of our paper as much as possible. If there is any omission in our response or if you have any additional question, please feel free to let us know. Thanks again for your great efforts!

---

> > ### Comment · Reviewer_iFKj · 2024-08-12
> >
> > I have read all the reviewers' comments and the authors' responses. My concerns have been addressed by the authors. I will not reduce my score.

---

> > > ### Author Response · Authors · 2024-08-12
> > >
> > > Dear Reviewer iFKj,
> > >
> > > We greatly appreciate your thoughtful comments and are pleased that you found our responses satisfactory. We will incorporate these important discussions to revise the final manuscript.
> > >
> > > Thank you once again for your valuable suggestions and feedback. We really enjoy discussing with you and sincerely appreciate your effors.

---

### Author Rebuttal · Authors · 2024-08-05

We thank all the reviewers, ACs, SACs and PCs for reviewing so many papers and providing insightful and objective opinions.

We are so grateful that:
- __All the reviewers give our work positive ratings (7556)__, which is a tremendous encouragement for us.
- Reviewer iFKj highly recommends our work for the novel and elegant formalization, the creative methodology and the solid experiments.
- Reviewer SGyo confirms the importance of our work, the insightful guidance strategy and the novel augmentation approach.
- Reviewer 7cPE appreciates the justifiable motivation, the exquisite illustrations and the practical value of our work.
- Reviewer BPQh highlights the creative insights, the detailed derivation and the comprehensive experiments of our work.

Meanwhile, all the reviewers offer valuable questions and suggestions. We have responsed to every reviewer individually in each separate rebuttal, accompanied by __a global rebuttal pdf__ of figures and tables down below. Please refer to the pdf as well.

We believe the rebuttal stage is highly meaningful, and we will add the rebuttal discussion to the new version of our paper as much as possible. If there is any omission in our response or if you have any additional question, please feel free to let us know. Thanks again for your great efforts!

---

### Decision · Program_Chairs · 2024-09-25

**Decision:**

Accept (poster)

**Comment:**

The work received positive scores from reviewers. They listed a number of strengths, including the practical importance of the problem, thorough derivations, and sufficient experiments. There were certain questions and concerns raised, which were adequately addressed during the discussion period. AC agrees with the reviewers, and believes that the problem to keep the subject while changing the prompts is indeed of utmost importance. The decision is to accept the manuscript.